**communications** engineering

# Real-time spatiotemporal optimization during imaging
Owen Dillon [1] ✉, Benjamin Lau[1], Shalini K. Vinod [2,3], Paul J. Keall[1], Tess Reynolds[1],
Jan-Jakob Sonke[4] & Ricky T. O'Brien[5]

High quality imaging is required for high quality medical care, especially in precision applications such as radiation therapy. Patient motion during image acquisition reduces image quality and is either accepted or dealt with retrospectively during image reconstruction. Here we formalize a general approach in which data acquisition is treated as a spatiotemporal optimization problem to solve in real time so that the acquired data has a specific structure that can be exploited during reconstruction. We provide results of the first-in-world clinical trial implementation of our spatiotemporal optimization approach, applied to respiratory correlated 4D cone beam computed tomography for lung cancer radiation therapy (NCT04070586, ethics approval 2019/ETH09968). Performing spatiotemporal optimization allowed us to maintain or improve image quality relative to the current clinical standard while reducing scan time by 63% and reducing scan radiation by 85%, improving clinical throughput and reducing the risk of secondary tumors. This result motivates application of the general spatiotemporal optimization approach to other types of patient motion such as cardiac signals and other modalities such as CT and MRI.

Patient motion during image acquisition is often unavoidable and can lead to reduced/inconsistent image quality[1–4]. For example, respiratory motion occurs during thoracic CT or MRI acquisition, leading to motion blur in the resulting image that can complicate diagnostic quality[5] or precision applications such as guiding a treatment beam to the tumor in radiation therapy[6]. Common strategies to remove patient respiratory motion from images include having the patient perform a breath-hold[7], retrospectively sorting data and only reconstructing data from a desired motion state (respiratory correlated)[8] or controlling hardware to only acquire data during the desired motion state typically based on a surrogate signal (prospective gating)[9–11]. Note that motion management methods such as correlated and gated imaging have analogies to other sources of patient motion e.g., cardiac[4,10,11], but may not be applicable to non-periodic motion e.g., bowel movements[3].

In some cases, high spatial resolution of anatomy and temporal resolution of underlying motion is desired e.g., in radiation therapy clinicians want to see the tumor boundary across the respiratory cycle to ensure the tumor is covered by the treatment beam as the patient breathes[6]. In such cases a 4D scan is performed, where the 4D image is composed of several 3D images each corresponding to a phase of the motion. A common example is 4DCT consisting of 10 3D images each corresponding to 10% steps through the respiratory cycle. Typical 4D protocols oversample during acquisition[6,8],

so that when data is retrospectively sorted by motion phase and independently reconstructed, the data for each individual phase is sufficient. This approach results in acquisitions that are slower than needed for most patients, and with higher scan dose in modalities using ionizing radiation[8]. The resulting image quality is also inconsistent between 3D frames of the 4D image due to the acquired data having uneven spatiotemporal distribution[12].

An alternative approach to oversampling with retrospective sorting is to prospectively control the acquisition hardware in response to measured patient motion. We call this approach Spatiotemporal Optimization (STO), with a high-level summary provided in Fig. 1. We can use STO to ensure a particular structure in the acquired data, for example that the data acquired in each motion phase is fully sampled but with minimal oversampling. A key consideration in STO as opposed to simple gating or the more sophisticated control in[11,12] is how the acquired data relates to the entire 4D image, so STO acquisition protocol and 4D reconstruction method should be selected to synergize. In the present work we make use of an "adaptive" 4D reconstruction method that estimates each 3D frame of the 4D image as perturbations of an initial motion blurred 3D image. The adaptive reconstruction method was able to maintain image quality as the total amount of acquired data was reduced thanks to the specific structure of STO data.

[1]University of Sydney, Faculty of Medicine and Health, Image X Institute, Sydney, Australia. [2]University of New South Wales, South Western Sydney Clinical School & Ingham Institute for Applied Medical Research, Sydney, Australia. [3]Liverpool Cancer Therapy Centre, Liverpool Hospital, Liverpool, Australia. [4]Department of Radiation Oncology, The Netherlands Cancer Institute, Amsterdam, The Netherlands. [5]Royal Melbourne Institute of Technology, School of Health and Biomedical Sciences, Medical Imaging Facility, Melbourne, Australia. ✉e-mail: owen.dillon@sydney.edu.au

**Fig. 1 | Overview comparing conventional and Spatiotemporal Optimization (STO) imaging for lung cancer radiation therapy guidance.** We use blue lines to represent angles at which peak inhale projections are acquired and yellow lines to indicate where peak exhale images are acquired, noting that the actual number of projections acquired is much higher and these are just to illustrate the concept. Conventionally acquired data has no consistent spatiotemporal structure, so typically a large amount of data is acquired to ensure acceptable image quality. In STO imaging the patient is continuously monitored and an optimization is performed to guide the imaging hardware so that the acquired data has a specific spatiotemporal structure. The spatiotemporal structure can be exploited during image reconstruction to produce images with high quality relative to the total amount of data acquired. Reconstructed inhale/exhale images of the alternative acquisition/reconstruction approaches shown along bottom row with patient consent.

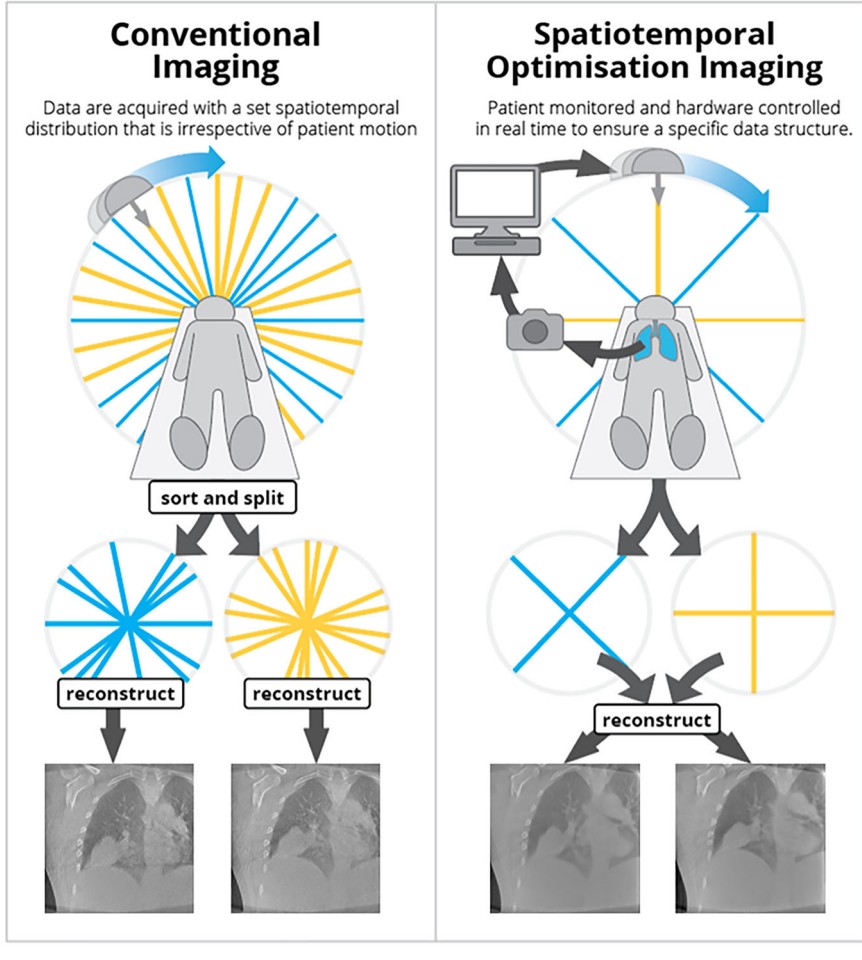

We present STO in this paper as a general framework for 4D imaging in which the data acquisition responds to a surrogate signal so that the reconstructed 4D image will have the best image quality relative to the scan time or total amount of acquired data. Sophisticated control in response to surrogate signals was implemented in[11] however in that case the focus was on isolating cardiac and respiratory motion in the acquired data so that the 3D reconstructed image had minimal motion blur. The phantom study presented in[12] acquires data in response to a respiratory surrogate signal based on a heuristic observation on the relationship between data structure and image quality with conventional 4D reconstruction. The simulation study presented in[13] developed reconstruction algorithms in response to structured data and demonstrated that certain 4D algorithms perform particularly well when data has certain structures.

In this manuscript, we describe the first implementation of Spatiotemporal Optimization in a clinical scenario and present the full cohort results of the first clinical trial. Our aim was to reduce scan time and scan dose while maintaining image quality, reducing radiation exposure to the patient, and improving clinical throughput without otherwise impacting the clinical workflow. We implement STO on the integrated imaging hardware of a radiation therapy linear accelerator (linac) to perform 4D Cone Beam Computed Tomography (CBCT) on 30 patients with lung cancer treated with radiation therapy during their first 2 treatment fractions. In lung cancer radiation therapy European practice guidelines recommend 4DCBCT[14] particularly when treating in fewer treatment sessions, as it is critical to ensure the tumor is well contained in the high dose region while avoiding radiosensitive organs as the patient breathes. Overview of the trial design is presented in Fig. 2. Patients receive a conventional 4D CBCT scan consisting of 1320 x-ray projections over a 240 s acquisition with conventional reconstruction[8]. The patient is then treated, followed by a pair of STO scans.

First, we acquired a 600 projection STO acquisition with conventional reconstruction (STO600) scan, followed by a 200 projection with adaptive reconstruction (STO200) scan. All projections are acquired at 120 kVp with 20 mA for 25 ms. Conventional reconstruction uses just the data acquired at each motion state to reconstruct each 3D frame of the 4D image separately, while adaptive reconstruction combines all the data in reconstruction each frame as described in Fig. 3. We compared scan time, dose, and image quality relative to the conventional 4DCBCT scan.

## Results

The ADAPT clinical trial (NCT04070586, ethics approval 2019/ETH09968) achieved STO in human patients for the first time. Key results are summarized in Table 1. The target dose reductions relative to the 1320 projection 240 s conventional acquisition were achieved, with the STO600 scan acquiring 55% fewer projections and the STO200 scan acquiring 85% fewer projections. The STO600 scan had a mean acquisition time of 242 s i.e., 1% increase in scan time. The shortest STO600 acquisition was 134 s, longest was 409 s, with 50% between 198 and 283 s. The STO200 scan had a mean acquisition time of 91 s i.e., a mean 64% reduction in scan time. The shortest STO200 scan was 43 seconds, the longest was 175 s with 50% between 71 to 108 sec.

### Implementation quality quantification

A key aspect of this first-in-world STO study was to show that monitoring the patient signal, solving the spatiotemporal optimization problem, and controlling the hardware to acquire data with a specific structure is clinically achievable. In this case, the desired data structure was even spread across both the angular arc and the respiratory cycle i.e., if a projection is acquired of respiratory bin $j$ the next projection is of respiratory phase $j + 1$, whereas

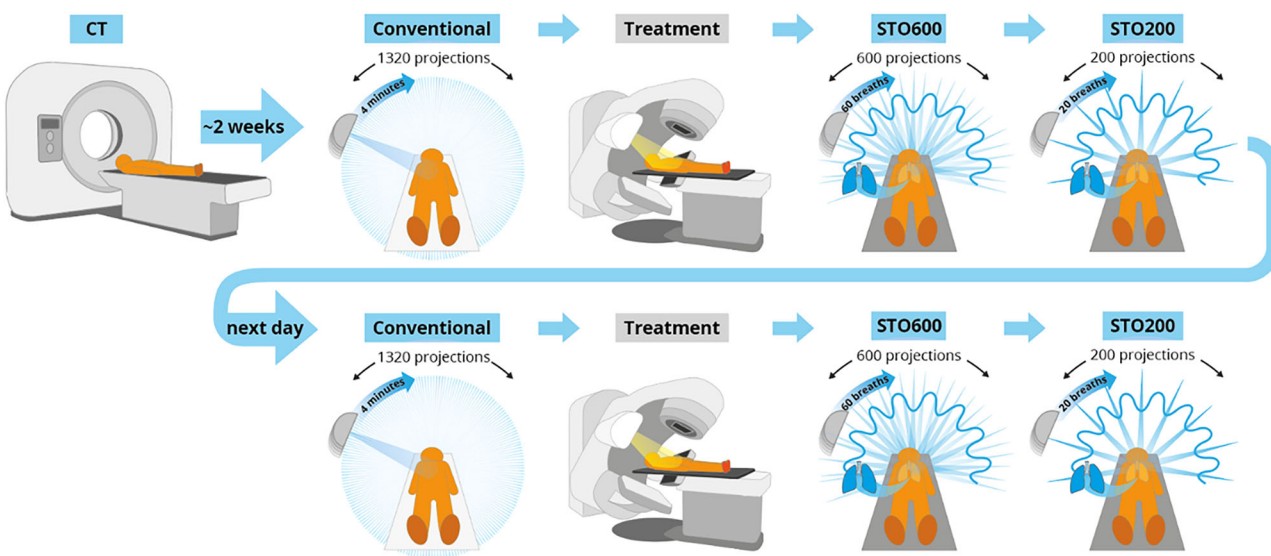

**Fig. 2 |** Overview of the trial design for the first application of Spatiotemporal Optimization (STO) imaging.

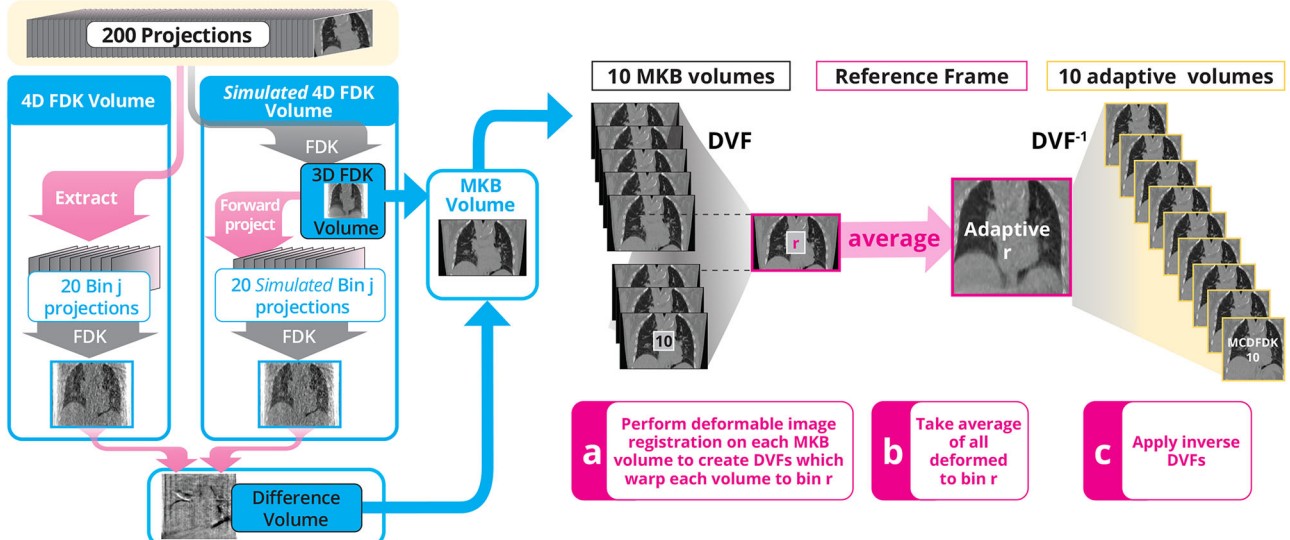

**Fig. 3 | Overview of the adaptive reconstruction algorithm used for the STO200 scan.** The procedure begins with a 4D Feldkamp-Davis-Kress (FDK) reconstruction. The 3DFDK volume is forward projected and then the 4DFDK algorithm used again to estimate perturbations, which then create the McKinnon-Bates (MKB) volumes. The motion between MKB volumes is estimated using Deformable Image Registration (DIR) to produce Deformation Vector Fields (DVFs) that are then applied and averaged to create the reference frame adaptive volume. Inverse DVFs are then applied to create the 4D adaptive image. Note that the algorithm requires 2 uses of the 4DFDK algorithm, a forward projection, and 9 deformable image registrations and 9 DVF inversions requiring a total of approximately 20 min of computation on our desktop hardware. Images provided with patient consent.

## Table 1 | Summary of the acquired scans and reconstructed image quality

| Summary of Scans, Acquisition and Image Quality Results | | | | | | | |
|---|---|---|---|---|---|---|---|
| Scan Name | Reconstruction Method | Number of Projections | Mean Scan Time (S) | Data Structure Accuracy (°) | CNR | TIW-T (mm) | TIW-D (mm) |
| Conventional | Conventional | 1320 | 240 | N/A | 7.5 | 7.8 | 7.7 |
| STO600 | Conventional | 600 | 242 | 0.71 | 5.9 | 10.2 | 9.4 |
| STO200 | Motion Compensated | 200 | 91 | 1.75 | 12.4 | 5 | 3.5 |

Note that Data structure accuracy, Contrast to Noise Ratio (CNR), Tissue Interface Width at the tumor (TIW-T) and diaphragm (TIW-D) are given as means across the entire dataset for each scan.

in a conventional scan often projections are acquired at the same respiratory phase with less than 0.2° angular difference. Across the patient cohort as shown in Fig. 4, the conventional acquisitions performed in this trial acquired approximately 34,000 largely redundant chest x-rays across the 30 patients[8,11]. The STO600 and STO200 scans were generally close to their desired interbin angular spread of 3.3° and 10° respectively. The STO600 scan seems to have a tighter angular spread, likely because the slower required gantry rotation made acquiring consistent data easier than in the faster STO200 scan. The STO200 acquisition also has some relatively large departures from the ideal spread, with some gaps over 15° relative to the target 10°. These departures stemmed from 2 scans where the patients had approximately 20 s periods of very irregular breathing, potentially due

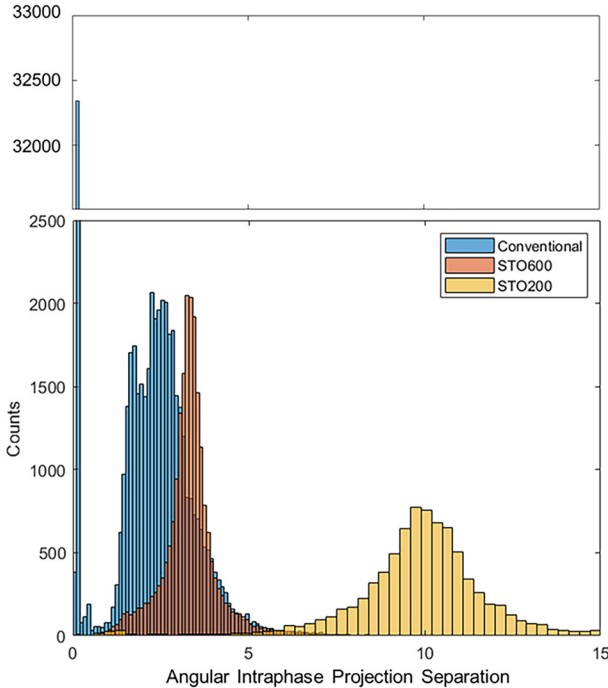

**Fig. 4 | Angular separation between projections of a given phase, across all scans.** Recall that the Spatiotemporal Optimization 600 projection (STO600) scan targets a 3.3° separation and the STO200 scan targets a 10° separation. Note the y axis break.

to coughing. The Mean Absolute Error (MAE)[12] relative to the STO target data structure for STO600 scans was 0.71° and the MAE for STO200 scans was 1.75° i.e., on average within 1° and 2° of the ideal, respectively.

## Image quality quantification

The supplementary materials to this article includes central slice tomographs for 5 selected cases. We present and discuss static images in the manuscript, but we encourage the reader to view the supplementary material and appendix available at https://github.com/Image-X-Institute/ADAPT-Example-Data to make their own qualitative assessment of the images. Note that all presented patient images are included as per the clinical trial NCT04070586 with ethics approval 2019/ETH09968 and patients provided written informed consent that such images may be used in publications.

Central slice coronal tomographs are shown in Fig. 5 for a conventional 4DCBCT (conventional acquisition, conventional reconstruction) as well as the STO600 and STO200 acquisitions with conventional and adaptive reconstruction. The STO600 acquisition conventional reconstruction scan has comparable image quality to conventional acquisition conventional reconstruction scan while reducing scan dose just by altering the acquisition. The STO200 acquisition adaptive reconstruction scan allows large reductions in scan time and dose while approximately maintaining image quality thanks to the adaptive reconstruction method working well with the STO data structure. The STO200 acquisition conventional reconstruction scans had clearly unacceptable image quality, and the STO600 acquisition adaptive reconstruction scan does not demonstrate a sufficient improvement in image quality over the STO200 adaptive reconstruction scan to justify the additional scan dose and time while still having the clinical concerns around use of motion compensated reconstruction. We have therefore omitted analysis of these scans, however more extensive investigations of the relationship between acquisition and reconstruction can be found in[13,15].

Compare the STO600 scan image to the conventional scan image in Fig. 5. There is a slight increase in image noise as expected from halving the imaging dose but no clear increase in artefacts. Comparing the STO200 scan image to the conventional scan image, image noise is reduced and anatomy is qualitatively more well defined.

Image quality metrics are shown in Fig. 6. The median Contrast to Noise Ratio (CNR)[16] for the conventional, STO600 and STO200 scans are 7.5, 5.9 and 12.4 respectively, suggesting that contrast is slightly reduced between conventional and STO600 images but improved in the STO200 images, which we observed qualitatively from Fig. 5. Paired-sample t-test of

**Fig. 5 | Central coronal slice tomographs for conventional 4DCBCT and Spatiotemporal Optimization (STO) imaging at different combinations of scan time and reconstruction algorithm.** The observed reduction in streaking in the adaptive reconstruction is due to the STO600 adaptive reconstruction being computed from 600 projections and the STO200 adaptive reconstruction being computed from 200 projections, around which point acquiring further projections leads to minimal improvement in filtered backprojection image quality. The conventional reconstruction images from the conventional, STO600 and STO200 are computed from 132, 60, and 20 projections respectively i.e. just the projections acquired at that phase. Images provided with patient consent.

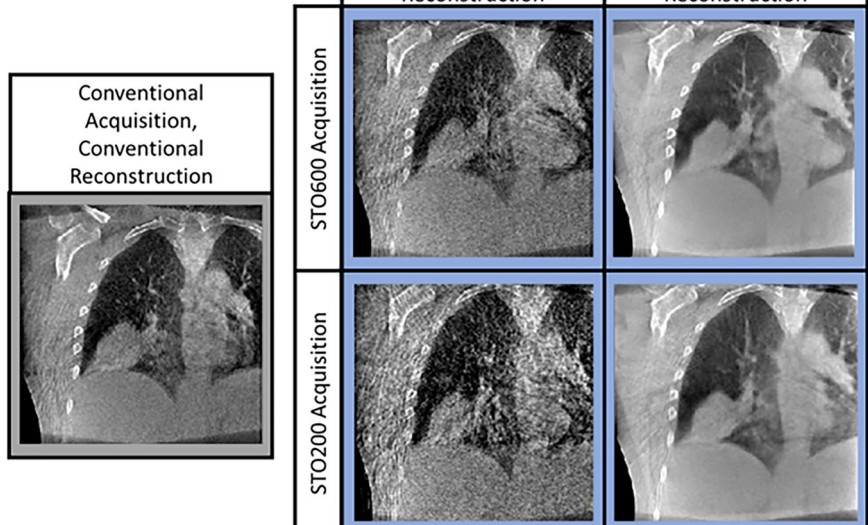

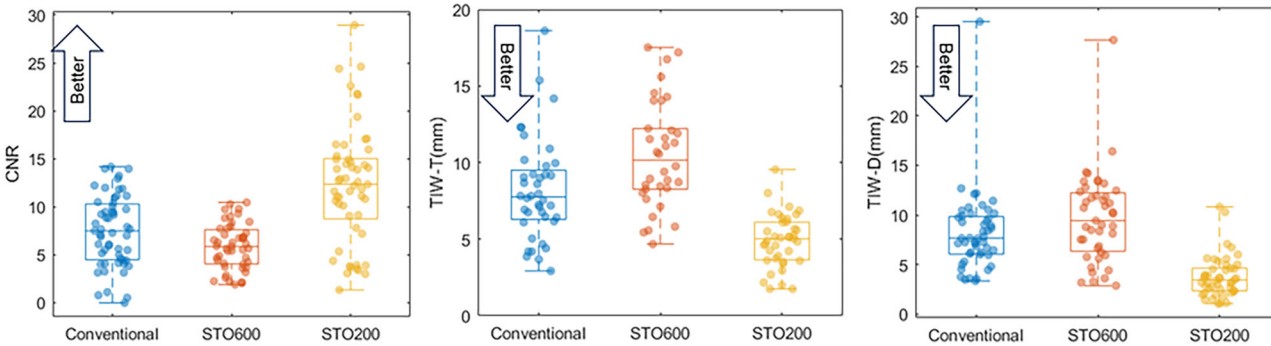

**Fig. 6 | Image quality metrics.** Note that "better" performance by each metric is higher CNR, lower TIW. Box plot centre line, box edges and whisker ends represent median, ±25% and ±50% (full range) respectively.

CNR values between conventional to STO600, conventional to STO200 and STO600 to STO200 gave $p$-values of $6 \times 10^{-9}$, $3 \times 10^{-9}$ and $1 \times 10^{-11}$ respectively. The median tumor Tissue Interface Width (TIW-T)[17] for conventional, STO600 and STO200 images were 7.8 mm, 10.2 mm and 5.0 mm respectively. This suggests the tumor boundary was slightly better defined in the STO200 images, which is a key concern when verifying that the tumor is being targeted effectively[6,18]. Paired-sample t-test of TIW-T values between conventional to STO600, conventional to STO200 and STO600 to STO200 gave $p$-values of $1 \times 10^{-4}$, $3 \times 10^{-7}$ and $6 \times 10^{-11}$ respectively. Similarly, the median diaphragm TIW (TIW-D) for conventional, STO600 and STO200 scans was 7.7 mm, 9.4 mm and 3.5 mm respectively, again suggesting that STO200 images have a more sharply defined diaphragm, which is the tissue interface with the largest respiratory motion amplitude in the lung. Paired-sample t-test of TIW-D values between conventional to STO600, conventional to STO200 and STO600 to STO200 gave p-values of 0.001, $4 \times 10^{-12}$ and $3 \times 10^{-10}$ respectively.

We measured motion model robustness[19] by deformably registering motion compensated STO200 images to the non-motion compensated STO600 images and extracting the motion in the tumor region as per[20]. We observed mean displacements of 0.28 mm, -0.0041 mm and 0.11 mm with standard deviation 0.70 mm, 0.75 mm and 0.68 mm in the superior/inferior, posterior/anterior and left/right axes respectively. The total displacements had mean of 1.2 mm and standard deviation 0.47 mm. These results suggest strong consistency in tumor positioning[19] well within the anatomically expected amount of variation over the acquisition time for both scans[21].

## Discussion
### Implementation quality
We successfully performed the first clinical trial using STO imaging, applied to 4DCBCT for lung cancer radiation therapy imaging. Relative to the conventional acquisition, the STO600 acquisition used 55% less imaging radiation, and the STO200 acquisition used 85% less imaging radiation. Reduction in scan time was more variable, going from a conventional scan of fixed length to STO600 scans taking 60 breaths and STO200 lasting 20 breaths. In this patient cohort the STO600 scan was on average 1% slower than the conventional scan, while the STO200 scan was 63% faster. So, while both STO600 and STO200 represent substantial reduction in scan dose[22], STO200 would be preferable if scan time was a significant clinical concern[22,23].

We found that conventional acquisition acquires projections at almost identical spatiotemporal location approximately 45% of the time, resulting in a substantial amount of scan time and/or dose that is contributing relatively little information[8,12]. In our implementation of STO, we were able to deliver accurate temporal spread, and were on average within 1° or 2° of the ideal spatial spread in the STO600 and STO200 scans respectively. The spatiotemporal evenness can be visualized in Fig. 4, where we can see STO solves the problem of acquiring redundant information, but hardware limitations of our implementation meant the target angular spread was not always achieved. The STO600 acquisitions were closer to the ideal than the

STO200 acquisitions, likely because the rotation steps were smaller and rotation speeds lower in the STO600 scan, so errors were relatively small. It may be that in new generation linacs and CBCT c-arm systems that are capable of much faster gantry rotations, a more precise STO implementation would be required to achieve scans sufficiently close to the desired data structure.

The data structure was sufficiently close to the ideal that we could get high quality reconstructions, but it may be that with other reconstruction algorithms or in other imaging modalities a more precise STO implementation will be required to achieve acquisitions sufficiently close to the desired data structure. It is worth noting when developing STO in other modalities, that the methods need a certain degree of robustness to variability in the data structure due to the realities of hardware control, patient variability and the ability to solve the spatiotemporal optimization problem in real-time.

### Image quality
Throughout this discussion we will refer not only to the quantitative metrics and figures provided within the manuscript, but also to features visible in the animations provided as supplementary material to this manuscript. Given the focus on 4D images in this study we encourage readers to view these supplementary materials and make their own qualitative assessment particularly with regards to the use of motion compensated reconstruction for the STO200 scan.

There was minimal visible difference between the conventional images and the STO600 images, and this is verified in the image quality metrics. We believe this is largely due to ~45% of projections in the conventional scan being acquired at almost identical spatiotemporal locations, and therefore not contributing much information that would improve reconstructed image quality. The STO600 scan acquires 55% fewer projections and ensures this "doubling up" of projections never occurs, so the resulting images are quite similar even though the same reconstruction algorithm is used. Because the standard reconstruction algorithm is used in the STO600 scans, the image artefacts are familiar to clinicians, and they may be more willing to adopt STO600 clinically purely for the scan dose reduction.

The STO200 images initially appear qualitatively clearer than conventional images, and some of the image quality metrics support this claim. This suggests that the presented STO method may be sufficiently robust even for the relatively large spatiotemporal variation in the data observed in 2 STO scans. While the STO200 scan uses mean 63% less scan time and 85% less scan dose while delivering quantifiably improved image quality, the use of deformable image registration to perform motion compensated reconstruction[24] may hinder clinical uptake. Deformable image registration when used clinically often involves a manual verification step, suggesting a degree of mistrust[25] and does have the potential to introduce artefacts in the final images. We invite the viewer to view the images in this manuscript and the animations in the supplementary material to make their own judgement if the motion compensation is sufficiently reliable for the clinical use case. In the case of adaptive reconstruction however, the final image is essentially a

weighted average of 10 deformable registration results so a single poor registration would visually manifest as a blurred piece of anatomy[13], which is visibly tractable. The relatively good TIW values for STO200 images at the diaphragm and tumor boundaries would suggest that representative motion is being captured, at least at these locations with the largest and most critical motion respectively.

As we are considering a 4D modality we direct the readers to review the supplementary materials to this manuscript which contains central slice tomographs of some representative patient scans. As observed in the static images the conventional and STO600 images are fairly similar albeit with some additional noise in the STO600 images likely due to acquiring around half as many projections. As in the static images we observe that the STO200 images have some widespread blurring likely due to imprecise motion compensation, however additional artefacts become noticeable in the animations. Most immediately apparent is a general unrealistic flexing in the images particularly near the periphery. This is most likely due to the image registration in this region aligning streaks to streaks rather than meaningfully capturing moving anatomy. In practice the images presented to physicians are masked so that only the part of the image well contained in the geometric field of view is visible, so these edge artefacts would not be visible, but we include them here as they may be considered of interest to the research community. We observe in the case 2 STO200 scan that the tumor moves erratically near peak exhale as registration of this phase may be inaccurate, and this is reflected by the additional blurring of the tumor edge in this case. More accurate registration or a synthetic middle phase as in[24] may reduce these effects. We also observe in case 3 that the STO200 tumor has little motion relative to the conventional scan image, however we similarly observe reduced motion in the STO600 scan that does not use motion compensation, suggesting the possibility that this patient was gradually reducing their breathing amplitude.

The key clinical utility of 4D imaging is visualization of where the patient anatomy moves during the respiratory cycle, so the 4D image must represent motion accurately. An advantage of only using filtered backprojection based reconstruction methods in this study is that any error in accounting for motion of a particular piece of anatomy will manifest as a blurred edge[13], so TIW can be used as a proxy for motion accuracy. There are concerns about motion model robustness when motion compensated reconstruction is used so we perform a similar analysis to[19] to confirm that the tumour location is consistent between the non-motion compensated STO600 scan and motion compensated STO200 scan acquired immediately after. The decision was made to compare STO600 and STO200 scans rather than back to the conventional as we do not expect the altered acquisition mode to impact patient motion but we do know patient motion changes over the ~20 min from conventional 4DCBCT acquisition through treatment delivery to STO acquisition[21]. We found that the tumour was within 1-2 voxels for each scan, confirming the motion model robustness considering the expected anatomic variation over the acquisition of both scans[21]. We show a wide range of images in the appendix and supplementary materials of this manuscript so that readers can assess the accuracy to which each scan represents motion and general clinical utility. An emerging technique in 4DCBCT is the use of dynamic reconstruction methods that reconstruct the anatomy at each projection without assuming reproducible motion[26–28]. Use of these reconstruction methods with STO acquired data may provide clinicians with a more complete view of the range of patient motion, acquiring data that represents each patient breath individually during the scan acquisition. In this study we chose to restrict to the periodic 4D scan paradigm as that is more clinically familiar, and the low TIW values suggest that the assumption was valid for this patient cohort.

Note that the STO200 image quality is comparable to 1,320 projection conventional scan image quality because STO data can be exploited by adaptive reconstruction even when the amount of acquired data is relatively low. In particular, the McKinnon-Bates (MKB)[29] step which estimates each phase as a perturbation of a motion blurred 3D image requires spatio-temporal evenness at this data sparsity, an observation echoed in[17,29]. The literature suggests spatiotemporal evenness is important for image quality in conventional and adaptive reconstruction[8,11,12,17,24,29] hence we targeted spatiotemporal evenness in this STO implementation.

There are a wide variety of alternative reconstruction algorithms for 4D CBCT beyond the conventional and adaptive explored here[30–32]. Note however that these algorithms do not suggest a particular spatiotemporal data structure that will yield best results, likely because without STO data structures cannot be enforced. It is possible that future work on 4DCBCT reconstruction will include spatiotemporal sensitivity analysis or assume spatiotemporal structure, now that STO has been demonstrated as viable in a clinical trial for the first time.

This work has been structured to demonstrate STO is performing in quantitative terms as much as possible e.g., by restricting to filtered backprojection type algorithms and analyzing tissue interface width, and by pursuing reductions in scan time and dose while quantitatively maintaining image quality so as to have minimal impact on the clinical decision making process. However, the key concern is whether the images are suitable for clinical use which necessarily involves a degree of qualitative assessment. Unfortunately, qualitative assessment by a panel of clinicians is a lengthy and expensive process and still subject to inter- and intra-observer variations. Another concern is that the qualitative difference in the images produced by each scan would complicate our ability to blind participants and introduce bias to the results. We therefore restrict this manuscript on the formalization and implementation of STO to a quantitative analysis as a necessary if not sufficient condition that the STO framework can maintain image quality relative to the amount of acquired data. Future work would involve qualitative investigation by clinicians, but the complexity and application specificity of such work would be more suitable for a discipline-specific journal. Readers can make their own qualitative assessment from images provided in the appendix and supplementary materials.

The results shown in this paper for STO in 4DCBCT motivates development of STO for other modalities. Work on achieving STO in 4DCT is underway[33,34] as analogies between 4DCT and 4DCBCT are clear. Note that the current 4DCT STO work does not suggest specific reconstruction methods tailored for STO data, however the use of filtered backprojection in CT suggests a similar perturbation type approach may give improved results. Extending STO to other modalities such as MRI may be more difficult as the relationship to 4DCBCT is less clear, however in any modality where acquisition is slow relative to the underlying cyclic motion[3], we believe STO is worth investigating e.g., synchronizing sampling of k-space to patient respiration to achieve desired data structure with respect to k-space and patient respiratory phase.

We have formalized a general method called spatiotemporal optimization (STO) for improving image quality relative to the amount of data collected by acquiring data with a specific spatiotemporal structure that can be exploited during reconstruction. We conducted the first clinical trial of STO in which the method was applied to imaging for lung cancer radiation therapy, and across 30 patients found that STO maintained image quality while enabling 63% reduction in mean scan time and 85% reduction in scan dose relative to the current clinical standard. This motivates implementation of STO for a wider range of modalities for example in CT and MRI and a wider range of patient motions for example cardiac[11,34].

## Methods

We first present STO as a general framework that can be extended to other modalities and types of motion. We then detail first in world clinical trial implementation of STO, on a radiation therapy imaging system. Next, we explain conventional and adaptive reconstruction, noting that the structure of respiratory STO data can be exploited by adaptive acquisition for lung imaging. We then detail the clinical trial design. Finally, we detail how we quantify performance of our STO implementation.

## Spatiotemporal Optimization Framework

A high-level overview of STO is provided in Fig. 1. The motivation for STO was to reduce scan time and dose while maintaining image quality, improving clinical throughput and patient safety respectively without otherwise impacting clinicians. We observed that image quality was a function not just of the total acquired data but the spatiotemporal structure of the data i.e., when, where and what data was acquired during the scan, and that different reconstruction algorithms could leverage different data structures more effectively. We concluded that if we could ensure data was acquired with a specific spatiotemporal structure that image quality could be preserved while reducing the total amount of data acquired.

The concept of STO can be formalized as follows. Let a general model for a system be $d = M(x, e)$ where $d$ is the measured data, $x$ is the unknown of interest, $e$ is noise and $M$ is the model relating the unknown to the measurements. Note that this general formulation is common in Inverse Problems and Design of Experiments literature which motivated development of STO. The reconstructed estimate of $x$ can be expressed as $\hat{x} = R(d) = R(M(x_{gt}, e_{gt}))$ where $R$ is the reconstruction method and we specify that the measured data reflects ground truth realisations of $x$ and $e$. We choose a reconstruction method such that $\psi(x_{gt}, \hat{x})$ is small where $\psi$ is a functional measuring similarity of $x_{gt}$ and $\hat{x}$ e.g., $\psi(x_{gt}, \hat{x}) = ||x_{gt} - \hat{x}||_2$ being a common example. We expand our measurement model to

$$d = M(x(t), e, S, T) \tag{1}$$

i.e., we allow $x$ to change with time $t$ and we specify that data is acquired at locations $S$ and at times $T$. We observe that reconstructed image quality is a function of $S$ and $T$ i.e., there exists some ideal choice

$$S_{ideal}, T_{ideal} = \min_{S,T} \psi\left(x_{gt}, R\left(M\left(x_{gt}(t), e_{gt}, S, T\right)\right)\right) \tag{2}$$

which may not be unique. It is in general not possible to find the above minimum, but we may find an approximation through simulation studies or choose $S_{ideal}, T_{ideal}$ according to some heuristic e.g., even spacing.

We usually don't know $x_{gt}(t)$, which impacts the choice of $S_{ideal}$ and $T_{ideal}$. We may however be able to monitor a surrogate signal $y(t)$ and pass a control signal $c(t)$ to the imaging hardware so that the ideal data is still collected. For example, surrogate signal $y(t)$ could be optically tracked position of a marker on the patients chest as a surrogate for respiratory state and $c(t)$ would be a signal telling the CBCT c-arm where to move and when to acquire a projection. More formally we define the surrogate signal as having the property that $y(t_1) = y(t_2)$ implies $x_{gt}(t_1) \approx x_{gt}(t_2)$. The control signal is used as $S(t) = S(t - \tau, c(t - \tau))$ and $T(t) = T(t - \tau, c(t - \tau))$ where $\tau$ is some system latency i.e., the control signal allows us to specify when and where the next data point will be acquired subject to some latency. We construct a model $P(t - \tau, y(t - \tau)) \approx y(t)$ that allows us to predict the surrogate signal. The spatiotemporal optimisation problem to be solved in real time becomes

$$c_{STO}(t - \tau) = \min_{c(t-\tau)} \phi\left(S_{ideal}\left(P(y(t - \tau)), c(t - \tau)\right), T_{ideal}\left(P(y(t - \tau)), c(t - \tau)\right)\right) \tag{3}$$

where $\phi$ is some measure of spatiotemporal similarity e.g., 2-norm. In other words, the real time spatiotemporal optimisation is to estimate a hardware control signal given the current patient surrogate signal such that the acquired data at some future time point has spatiotemporal structure as near as possible to what produces the best reconstructed image. Note that the improvement in image quality from STO depends on the accuracy of the surrogate signal and its prediction, the precision of our control over the hardware, and the sensitivity of the reconstruction method to altered spatiotemporal structure.

## First STO implementation: X-ray imaging in lung cancer radiation therapy

With the framework defined we can now describe the specific modality we chose for the first implementation of STO. We chose to implement STO for 4D Cone-Beam Computed Tomography (4DCBCT), in which a kilovoltage (kV) x-ray source facing a Flat Panel Detector (FPD) are rotated around a patient acquiring 2D x-ray projections from various gantry angles as the patient breathes[8]. The projections are used to reconstruct several 3D images each corresponding to a "bin" or portion of the patient's respiratory cycle, the set of which form the 4D image. It is common to acquire a 4DCBCT image on a radiation therapy linear accelerator (linac) immediately before delivering lung cancer radiation therapy, to ensure that the treatment will be delivered to the target while sparing healthy tissue while the patient breathes. Note that in this application of STO, the spatial variable $S$ is one dimensional and refers to the angle $\theta$ of the gantry. The temporal variable $T$ refers to the time at which a single projection image is acquired.

The conventional 4DCBCT scan in standard clinical practice acquires 1,320 2D x-ray projections over a $\theta_{arc} = 200°$ arc and $\Delta_{arc} = 240$ seconds, a significant amount of time given that the treatment room may be booked for as little as 10 minutes per patient. The data has an even spatiotemporal distribution with respect to gantry angle location and timing at which each projection is acquired, however this results in an uneven spread relative to the patient's respiratory cycle[8,12]. Unless the patient breaths very fast there is highly redundant data i.e., projections acquired at almost the same gantry angle of a patient at almost the same respiratory state. Data can also be sparse e.g., if a patient holds their breath near inhale, that temporal state is well sampled but there are large spatial gaps in data of other respiratory states. Reconstructions produced from data that had uneven spatiotemporal distribution relative to the patient's respiratory cycle were found to have reduced image quality. This observation motivated development of STO imaging.

It is standard in 4DCBCT to make the approximation that $x_{gt}(t) \approx x_j$ for all $t \in \Gamma_j$ where $1 \leq j \leq N_b$ is the "motion bin" separated into $N_b$ respiratory states and $\Gamma_j$ are the times at which the patient is in that respiratory state. The reconstruction problem becomes estimating each $N_b$ 3D image $x_j$ rather trying to solve for all $x_{gt}(t)$. Respiratory motion is taken to be cyclic so $j + N_b = j$ and at bin $j = N_b$ we define the following bin as $j + 1 = 1$. We define $t_k$ to be the time at which a projection is acquired with $1 \leq k \leq N_p$ where $N_p$ the total number of projections acquired in the scan i.e., $N_{p,conv} = 1,320$ for a conventional 4DCBCT acquisition. The spatiotemporal distribution of the conventional acquisition can be expressed as

$$S_{conv}(t_{k+1}) - S_{conv}(t_k) = S_{conv}(t_{k+2}) - S_{conv}(t_{k+1}) = \theta_{conv} \text{ for all } k \tag{4}$$

$$T_{conv}(t_{k+1}) - T_{conv}(t_k) = T_{conv}(t_{k+2}) - T_{conv}(t_{k+1}) = \delta_{conv} \text{ for all } k \tag{5}$$

where $\theta_{conv} = \theta_{arc}/N_{p,conv}$ and $\delta_{conv} = \Delta_{arc}/N_{p,conv}$ i.e., even separation in space and time but with no reference to underlying patient state.

Our target spatiotemporal distribution $S_{ideal}, T_{ideal}$ was chosen heuristically to be even separation in terms of gantry angle and patient respiratory cycle[12]. We can express our target of spatiotemporal even separation with respect to patient respiration as

$$S_{ideal}(t_{k+1}) - S_{ideal}(t_k) = S_{ideal}(t_{k+2}) - S_{ideal}(t_{k+1}) = \theta_{ideal} \text{ for all } k \tag{6}$$

$$t_k \in \Gamma_j \text{ if and only if } t_{k+N_b} \in \Gamma_j \text{ and } t_k \in \Gamma_j \text{ if and only if } t_{k+1} \in \Gamma_{j+1} \text{ for all } j, k \tag{7}$$

Where $\theta_{ideal} = \theta_{arc}/N_{p,STO}$ as in Eq. 4 i.e., the spatial distribution remains an even spread as in the conventional acquisition. The temporal structure is defined in Eq. 7 where we impose that each subsequent projection corresponds to the subsequent motion state and make use of the motion being

periodic. Note that the combination of Eqs. 6 and 7 implies that

$$S_{ideal}\left(t_{k+N_b}\right) - S_{ideal}\left(t_k\right) = N_b\theta_{ideal} = \frac{N_b\theta_{arc}}{N_{p,STO}} \quad (8)$$

i.e., the spatial separation is even between projections and between projections of the same motion state. Our target functional $\psi$ in Eq. (3) was simply absolute difference relative to the ideal.

We placed a depth sensing camera on a tripod by the treatment couch pointed at the patient's chest, where the chest height was taken as the surrogate signal $y(t)$. Use of chest height as a surrogate for respiratory state is common in 4DCT and 4DCBCT[1,8]. The predictive model $P\left(y(t - \tau)\right) \approx y(t)$ was based on[12] and elaborated further in[35,36]. The predictive method fits an ellipse to a plot of $y(t)$ against $y(t - \omega)$ where $\omega$ is a variable lag we chose as 0.5 s. The plot traces an ellipse due to the sinusoidal motion, centred at the baseline drift in the surrogate signal. We divide the ellipse into $N_b$ segments and take the current segment currently traced from the major axis of the ellipse as being "phase" $j$, the $j$'th motion state. We predict the state at $t + \tau$ from the fitted ellipse.

We used external circuitry electronically isolated from the radiation therapy system to input the control signal $c(t)$. Gantry angle was monitored by an inclinometer attached to the gantry. The radiation therapy system control software was instructed to perform a fixed $6s^{-1}$ gantry rotation speed and 5.5 Hz projection acquisition, but we installed a potentiometer in the gantry speed control signal to modulate from $0s^{-1}$ to $6s^{-1}$ and a solid-state relay to supress unwanted projections. The control signal is the potentiometer and relay settings. Note the constraints on the control signal are (1) the gantry rotates in one direction (2) the gantry speed is no greater than $6s^{-1}$ (3) the scan would automatically abort if the gantry speed went below $0.3s^{-1}$ (4) solution for gantry acceleration must be below $2s^{-2}$ as that was the highest our system could achieve (5) we can only acquire or block projection acquisition at the 5.5 Hz rate.

The depth sensing camera signal is passed to a control computer that performs the surrogate signal prediction $P\left(y(t)\right) \approx y(t + \tau)$. The control computer then solves the optimisation problem in Eq. (3) to find the STO control signal. The STO control signal is then passed to the potentiometer control and solid-state relay. Solving the optimisation problem in Eq. (3) subject to our constraints on the control signal required use of a mixed integer quadratic programming (MIQP) technique described in detail in[12]. Our implementation made use of an approximate heuristic solution method described in[35].

Note that in our implementation of STO we added equipment to the clinical environment namely the camera for surrogate signal monitoring, the control electronics and the control computer. However, a commercial implementation would not require separate control electronics and control computer, and the surrogate signal could be obtained by integrating the depth sensing camera into the treatment room or use current commercial surrogate monitoring solutions such as the Varian RPM. Beyond that there is no modification to the clinical environment or workflow due to implementing STO imaging in this context.

Based on earlier simulation and phantom studies[12,13,35] we chose to perform 600 projection STO acquisition and 200 projection STO scan protocols, referred to as STO600 and STO200 respectively. In the simulation studies STO600 data with standard reconstruction and STO200 with our novel adaptive reconstruction method maintained image quality relative to the current clinical standard, and the aim of this first in patient clinical trial was to maintain image quality with the greatest possible reductions in scan time and dose. Further explanation of why the STO600 scan was performed is provided in the discussion. Note that the STO600 scan lasts 60 breaths and acquires 55% fewer projections so approximately 55% lower dose and the STO200 scan lasts 20 breaths with approximately 85% lower dose relative to the conventional 1320 projection 240 s conventional scan.

## Conventional and adaptive reconstruction

The standard algorithm used in radiation therapy imaging is a variation of filtered back projection adapted to cone beam geometry known as the Feldkamp-Davis-Kress (FDK) reconstruction algorithm[37]. The algorithm can be expressed as

$$\hat{x}_{FDK} = FA^T d \quad (9)$$

Where $x$ denotes the 3D patient voxel image, $\hat{x}_{FDK}$ is the FDK estimate, $F$ is the filter, $A$ the forward projection matrix, $A^T$ the back projection matrix and $d$ is the acquired data. Note that the FDK reconstruction algorithm involves 1 evaluation of $A^T$ and 1 evaluation of a filter.

The 4DFDK algorithm[8] can be stated as

$$\hat{x}_{4DFDK,j} = FA_j^T d_j \quad (10)$$

Where $1 \leq j \leq N_b$ refers to the $j$'th respiratory motion bin, $d_j$ is the data acquired of that state and $A_j$ is the corresponding forward projection matrix. Note that for conventional acquisition $d_j \approx 1/N_b$ of the data but with our STO target $d_j$ is exactly $1/N_b$ of the acquired data. Note that $\hat{x}_{FDK} = \sum_{j=1}^{N_b}\hat{x}_{4DFDK,j}$. Because each 3D frame of the 4DFDK image is reconstructed with much less data than the 3DFDK, each 4DFDK frame typically contains more noise than the 3DFDK image, however the 4DFDK frames have much less motion blur than the 3DFDK image. Note that reconstructing all 4DFDK frames involves evaluation of each $A_j^T$ but add to equivalent of 1 evaluation of $A^T$. The 4DFDK algorithm as stated involves $N_b$ evaluations of $F$ however implementation as back projection filtered (filtering of $d$) makes the total computation of the 4DFDK and 3DFDK algorithms roughly equivalent.

We developed an adaptive reconstruction algorithm that makes use of the computational efficiency of filtered backprojection, the similarity between each 4D frame, and motion compensation. An overview of the adaptive reconstruction method is shown in Fig. 3 and we provide a more detailed description below.

The adaptive reconstruction starts with a perturbation type approach known as the McKinnon-Bates (MKB) algorithm[29]. Information is shared between 4D frames by estimating each as a perturbation of the 3DFDK image. The 4DFDK is computed, added to form the 3DFDK and then forward projected to make simulated projections

$$d_{s,j} = A_j\hat{x}_{FDK} \quad (11)$$

corresponding to the 4DFDK projection binning, where subscript $s$ refers to simulated from the 3DFDK. The simulated projections are reconstructed as

$$\hat{x}_{s,j} = FA_j^T d_{s,j} \quad (12)$$

and affine scaled to match the 4DFDK images as

$$\left(\hat{a}, \hat{b}\right) = \min_{a,b}\left\{||a + b\hat{x}_{s,j} - \hat{x}_{4DFDK,j}||_2^2\right\} \quad (13)$$

which in turn are used to produce different images

$$\hat{x}_{diff,j} = \hat{a} + \hat{b}x_{s,j} - \hat{x}_{4DFDK,j} \quad (14)$$

which are the perturbations used to form each MKB frame

$$\hat{x}_{MKB,j} = \hat{x}_{3DFDK} - \hat{x}_{diff,j} \quad (15)$$

hence the description of MKB as a perturbation-based approach. Note that computation of MKB involves 2 evaluations of $A^T$, 1 evaluation of $A$, and 2

applications of filtering. The computational cost of MKB is therefore roughly 3-4 times the computational cost of 3DFDK or 4DFDK.

The adaptive reconstruction algorithm uses MKB images as an intermediate step for motion estimation, before performing motion compensation. Motion compensation for CBCT was introduced[24] as the Motion Compensated FDK (MCFDK) algorithm, where motion was estimated from the planning 4DCT image. Because patient motion changes between planning and treatment, alternative approaches such as MoCo[38] were developed in which motion is estimated from 4DFDK images. We found that the MoCo algorithm was not sufficient for the STO200 scan as the intermediate 4DFDK images were too noisy for reliable motion estimation. We instead estimate motion on the MKB images as

$$\hat{V}_j = \min_V M\left(\hat{x}_{MKB,r}, W\left(\hat{x}_{MKB,j}, V\right)\right) \qquad (16)$$

Where $\hat{V}_j$ is the estimated Deformation Vector Field (DVF) used by the function $W$ to warp the phase $j$ MKB image to match the reference phase $r$ MKB image, with the matching $M$ quantified as Mattes Mutual Information[39]. In our application phase $r$ was chosen to correspond to peak inhale.

The deformable image registration was performed using a B-splines algorithm implemented in the Elastix toolkit[40]. A relatively coarse grid was used, with 16 mm between control points. This was done to (1) reduce computation time, each registration taking 2 minutes on our hardware and (2) the underlying image quality is not good enough to justify performing a more precise registration. We found that this coarse grid was still sufficient to produce adequate results.

The adaptive phase $r$ image is calculated as

$$\hat{x}_{adaptive,r} = \sum_{j=1}^{N_b} W\left(\hat{x}_{MKB,j}, \hat{V}_j\right) \qquad (17)$$

noting that $W\left(\hat{x}_{MKB,r}, \hat{V}_r\right) = \hat{x}_{MKB,r}$. For computational efficiency the other phases are calculated as

$$\hat{x}_{adaptive,j} = W\left(\hat{x}_{adaptive,r}, \hat{V}_j^{-1}\right) \qquad (18)$$

as inverting the DVFs is faster than computing additional registrations.

Reconstructions were performed on a Dell workstation with Intel Xeon E5-2687 32 core 3.1 GHz CPU, NVIDIA RTX 8000 48GB VRAM 4608 CUDA core GPU, and 64GB of RAM. The FDK implementation was used from the Reconstruction Toolkit (RTK)[41] and registration performed using the Elastix toolkit[40]. In our implementation 4DFDK reconstruction took approximately 1 minute and MKB reconstruction took approximately 2 min. The 9 registrations for adaptive reconstruction added approximately 18 min so in total the adaptive reconstruction algorithm required 20 min however with sufficient computing the registrations could be performed in parallel to reduce computation time to 4–5 min using the current research implementation of our software. We believe that refining the software to avoid unnecessary read/write overheads, making use of hardware acceleration and particularly the integration of AI registration methods for inference[42] the computation time could be reduced to under half a minute.

Note that examples of the code used to perform the reconstructions in this study have been hosted online at https://github.com/Image-X-Institute/ADAPT-Example-Data.

## Clinical trial design

We conducted the first clinical trial of STO in the specific context of accounting for respiratory motion in cone beam computed tomography for lung cancer radiation therapy. The 30-patient trial was named ADaptive CT Acquisition for Personalised Thoracic imaging (ADAPT) (NCT04070586, ethics approval 2019/ETH09968). All participants provided written

informed consent to take part in the study. The trial was designed for a 30 patient cohort as this was predicted to give relatively strong statistical power and could be accrued within a year. The aim of the trial was to reduce scan time and dose without compromising image quality relative to the current standard, as this has the clear clinical benefit of reducing patient radiation exposure and improving clinical throughput without impacting how clinicians might use the images.

We chose to acquire 3 types of scan (1) a conventional 1320 projection 240 s scan as a baseline (2) a 600 projection over 60 breath STO scan (STO600) and (3) a 200 projection over 20 breaths (STO200) scan. We chose to acquire STO600 and STO200 scans as we found in simulations that STO600 acquisition conventional reconstruction and STO200 acquisition adaptive reconstruction scans had comparable image quality to conventional scans and wanted to verify this observation in practice. Implications of such a result are described in the discussion section.

The trial design is summarized in Fig. 2. Patients first receive a 4DCT scan for treatment planning. Approximately 2 weeks later, the patient comes to the linac bunker for their first treatment fraction. A conventional 4DCBCT scan is performed, acquiring 1320 projection images over 240 s. The patient is then treated as normal. We then acquire the STO600 scan, and then the STO200 scan. The following day the patient comes for their second treatment fraction, and we again acquire a conventional scan, deliver treatment, then acquire a STO600 and a STO200 scan. While we planned to acquire 4 STO scans for each of the 30 enrolled patients for 120 total STO scans, we cancelled scans if the patient reported discomfort so only 102 STO scans were acquired.

The trial cohort was restricted to lung cancer patients over 18 years old who could give informed consent and were to receive at least two treatment fractions and at least one conventional 4DCBCT scan. Patients were excluded if they were pregnant or if the treating physician thought they could not tolerate the extra scans. All patients were imaged and treated using an Elekta Synergy linac.

## Quantification of Implementation and Imaging Accuracy

The STO implementation aims to acquire data with a specific structure, in this case even spatially with respect to the imaging gantry and temporally with respect to patient respiratory phase. The data structure was selected early in the trial design based on simulation studies that indicated image quality would be quantitatively similar or better[13]. Methods of quantifying implementation accuracy were developed closer to patient accrual. The STO hardware recorded the patient phase, scan time, and times at which projections were acquired, allowing us to retrospectively assess the implementation accuracy. The scan time was recorded, and the reductions assessed against the conventional scan. Observing that the imaging hardware uses a fixed mAs and kV per projection and each scan is acquired over the same arc, percentage reduction in number of projections acquired was taken as a leading order approximation of the percentage reduction in scan radiation[18].

We quantify the STO acquisition accuracy with Mean Absolute Error (MAE)[11] relative to an idealized scan as $MAE = \sum_{j=1}^{N_p} \frac{1}{N_p} |S_{ideal,j} - S_{acquired,j}|$ where $N_p$ is the number of projections, $S_{ideal,j}$ is the ideal gantry angle of the $j$'th acquired projection and $S_{acquired,j}$ is the gantry angle of the actually acquired projection i.e. for the STO200 scan $N_p = 200$ and $S_{ideal,j} = j\theta_{arc}/N_p = j°$.

The aim of this study is to establish the reduction in scan time and dose that can be achieved with STO while still maintaining usable image quality. With that in mind, image quality is assessed relative to the conventional scan images.

To quantify how well tissue types can be distinguished, we compute the Contrast to Noise Ratio (CNR)[43] between the diaphragm and a homogeneous lung region as $CNR = \frac{\mu_{lung} - \mu_{diaphragm}}{\sigma}$ where $\mu_{lung}$ and $\mu_{diaphragm}$ are voxel value averages of regions of lung and diaphragm and $\sigma$ the joint standard deviation.

We quantify edge definition using Tissue Interface Width (TIW)[16] at the edge of the tumor (TIW-T) and diaphragm (TIW-D) and in cases that one of these tissues could not be clearly seen e.g., diaphragm out of frame, this metric was not calculated. A $5 \times 5 \times 60$ voxel region is placed at the tissue interface to get 25 runs of 60 voxels $v_j$. A sigmoid is fitted to each run, and the average width over which the sigmoids change from 90% to 10% is taken as the TIW. Similar methods have been used for image quality quantification in motion compensated 4DCBCT[16] and 4DCT[44].

A concern in motion compensated imaging is whether the motion model is robust, particularly in the tumor region. A method of testing motion model robustness proposed in[20] is to contour the tumor in images reconstructed with and without motion compensation and compare the centroid location. We implemented a more automated approach in which the motion compensated STO200 images are deformably registered to the STO600 images and the DVF values in the tumor are taken as a volumetric displacement as described in[19]. We compute a mean displacement in each axis as well as a total distance and compute standard deviations.

## Statistics and reproducibility

The underlying data and analysis codes are hosted online at https://github.com/Image-X-Institute/ADAPT-Example-Data.

## Reporting summary

Further information on research design is available in the Nature Portfolio Reporting Summary linked to this article.

## Data availability

Summary data and analysis codes are hosted online at https://github.com/Image-X-Institute/ADAPT-Example-Data. Raw imaging data of the clinical trial is not available as per trial protocol NCT04070586, ethics approval 2019/ETH09968.

## Code availability

Example codes are hosted online at https://github.com/Image-X-Institute/ADAPT-Example-Data.

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

## Acknowledgements

The authors would like to acknowledge contributions from Helen Ball for contributing to discussions and assisting in editing the manuscript. This work was supported by NHMRC project grant 1138899.

## Author contributions

O.D., R.O., P.K., and T.R. contributed to the initial concept. O.D. and R.O. wrote the experimental control, reconstruction, and analysis software. O.D., P.K. and R.O. developed the experiment design. O.D. and R.O. developed the mathematical formulation. B.L. contributed to data analysis. J-J.S. contributed to practical implementation, S.V. contributed extensively to the clinical trial. All authors reviewed the manuscript.

## Competing interests

O.D., R.O., B.K.F.L., and P.K. are in a non-exclusive licensing agreement of parts of the presented technology.
