## [Transparent Peer Review file · Communications Engineering]

Real-Time Spatiotemporal Optimization During Imaging

Corresponding Author: Dr Owen Dillon

Version 0:

Reviewer comments:

Reviewer #1

(Remarks to the Author)

Major claims: Scan time and scan dose are reduced using STO during 4D-CBCT while maintaining image quality and providing minimal impact on the clinical workflow. I believe all of these claims are backed, but if you claim to limit impact to the clinical workflow, I believe there should be some comments regarding the practical implications. Are there additional training, new equipment, additional setup or processing time, a need for therapists to interpret the successful or unsuccessful structure during acquisition, etc. It doesn't need to be too thorough, but there should be an acknowledgement of the impact on daily workflow and throughput other than reduced scan time and technical considerations.

Novelty: The approach is quite novel and impactful, so this paper describing the first results on patients in a clinical trial is very important, in my opinion. If other clinics implement this approach, they can achieve dose-efficient, and perhaps more importantly, time-efficient 4D-CBCTs with maintained image quality. This would be hugely beneficial.

Statistical Analysis: I think some key metrics are employed to characterize the important aspects of these imaging techniques: MAE to capture optimized angular distribution, SSIM to capture anatomical structures, CNR to capture image quality, and TIW to capture image sharpness. Summary statistics and paired t-tests are used to interpret. Based on the spread of data in Figure 6, I think the t-tests are appropriate.

Major comments:

1. This is up to the authors, but might you want to spend some time in the introduction describing the literature behind STO, and building up to the clinical trial?
2. The first paragraph of the results would be greatly aided by a table. It would just make it faster and easier for readers to compare results across acquisitions/reconstructions.
3. There are two scan sessions acquired for many patients, resulting in 102 scans. However, there seems to be less than 102 data points in Figure 6. Were some removed? Maybe it is just hard to see.
4. I think you could expand Figures 5 and 6 to include both acquisitions and reconstructions to better show off the impact of the adaptive reconstruction alone. At worst, you could expand Figure 5 to include these and leave Figure 6 alone. I fully understand the justification for highlighting STO600 conventional and STO200 adaptive, but still think Figure 5 is a good opportunity to further illustrate the direct impact of the reconstruction technique.

Minor comments:

1. Author affiliation 3 is unassigned.
2. There are a few run-on sentences that could use commas or breaks, specifically in lines 23, 44, 323, 439, 557.
3. Lines 103-105: You already introduced the shorthand for STO600 and STO200, so you can shorten these.
4. Line 109: First sentence sounds awkward.
5. Line 336: Remove "just"
6. Lines 118-120: If the t-test was significant, instead of saying CNR was "comparable" for STO600, I'd maybe say "slightly reduced."
7. Line 309: (4) not really worded as a constraint in this list of constraints.

8. Line 392: Capitalize “acquisition”?

9. Line 443: You could also cite <https://doi.org/10.1002/mp.12153> as I believe it is a very similar method.

10. Great job on all figures – each one is presented well and greatly enhances understanding.

Reviewer #2

(Remarks to the Author)

This paper presents Spatiotemporal Optimisation (STO) Imaging, a method of actively controlling the acquisition of medical imaging data during free breathing such that the data has a desired spatial-temporal structure. The paper presents this as a general approach but focusses on applying it to and evaluating it for 4DCBCT imaging. The paper also presents a complementary adaptive reconstruction algorithm that can be used together with the STO acquisition and provides improved images, especially when the number of projections is at its lowest (200). Finally, the paper also presents a clinical trial that performed STO acquisitions on 30 lung cancer patients and analyses the results.

Overall, the paper is well written and easy to follow, although I do find putting the methods section last, after the results and discussion, a confusing way of structuring a paper (but presume this is the format requested by the journal – if not I suggest changing it). In general, the presented methods seem sensible (although I do have some questions and comments on them – see below), and the clinical trial has collected an impressive amount of data in order to evaluate them. However, I think the main weakness of the paper currently is the analysis of the results – currently this does not provide very compelling evidence that the method provides images that are suitable for clinical use, and I think it would benefit from improvements in this area before publication. Below I provide a number of specific comments and suggestions on how the paper could be improved – note, I appreciate that the paper already contains an impressive amount of content, with presenting the STO method, the adaptive reconstruction, and the clinical trial to evaluate these, so including all of my suggestions may be beyond the scope of the current paper.

One important aspect currently missing from the analysis is that of the motion seen in the 4DCBCT scans – the whole point of 4D (as opposed to 3D) scans is to see how the tumour and other anatomy moves, so a critical question is whether the motion appears the same in the STO scans as it does in the conventional scans? This is especially pertinent for the adaptive reconstructions, where errors in the registrations will impact the perceived motion. I presume the SSIM is calculated for all phase images, so will be impacted by errors in estimating the motion, but it is also impacted by many other factors, and I think separate analysis of the ability of the STO scans to correctly estimate the motion is warranted. However, I appreciate that finding good ways to do this quantitatively is challenging (which leads to my next comment).

While I agree that it is necessary to include quantitative analysis as you have, I also think that qualitative analysis extremely valuable for assessing the quality of reconstructed images (and estimated motion). I would like to see many more examples of the reconstructed images from all methods, including the best and worse cases, as well as more typical examples – if there is not space to include these in the main paper they could be an appendix of supplementary material (although I think the main paper needs at least 2-3 examples to get any feel for how well the different methods have worked, e.g. based on the one example you have shown I question your assertion that the ST600-conv-recon is comparable to the conventional acquisition). Likewise, movies of the results should be made available either as supplementary material (if allowed) or via a link to download them, as they will be very useful for assessing how well the STO methods can reproduce both the image quality and motion seen in the conventional images. The qualitative evaluation could be made more formal and semi-quantitative by getting clinical staff to rate the images, which could add valuable evidence as to whether the images are suitable for replacing conventional scans clinically. It would also be useful to link the quantitative and qualitative evaluations, e.g. state the quantitative metrics for the cases shown and/or show the images for the cases with the best/worst metrics.

Related to the above, it would be interesting to include a brief discussion on exactly how the 4DCBCT images are used clinically, both now and how they may be used in the future (e.g. for auto-segmentation, for dose calculations and warping), as this will influence what is considered clinically acceptable image quality.

I think it would be interesting to further analyse the results for STO600-adapt-recon and STO200-conv-recon – while STO200-conv-recon is clearly worse than STO600-conv-recon visually, it would be informative to see how the metrics differed for all 4 images, and would help to interpret the metric results for the STO600-conv-recon and STO200-adapt-recon that have been provided. It would also help to assess the separate effects of varying the number of projections and the reconstruction method on the image quality and the estimated motion. Furthermore, from the one example shown it looks like STO600-adapt-recon may even give better results than the conventional imaging (which if it is correcting for the motion well it should do) – although this is very challenging to show when you consider the conventional images as the gold standard you are comparing to.

The STO method is well suited to addressing a specific problem with conventional 4DCBCT, that the projections are unevenly sampled (relative to phase), and hence many of them provide limited information. However, this is not the only potential problem/limitation with 4DCBCT. Another is that it, like many 4D imaging methods, assumes reproducible motion from one breath to another, and that the motion is strongly related to the surrogate signal used to measure the respiratory phase – and these assumptions do not always hold. While this study does not aim to address these issues, it does provide an interesting opportunity to study their potential impact on your proposed methods, especially as you noted that there were some individuals/scans that experienced periods of very irregular breathing. It would be good to discuss these cases in a bit more detail and in relation to the issue of irregular motion, and to show the images for these cases. Related to this, there have recently been some published methods that attempt to model the variable motion throughout the acquisition and do not rely on the assumption of reproducible motion, e.g.

Jailin, C., Roux, S., Sarrut, D. and Rit, S., 2021. Projection-based dynamic tomography. *Physics in Medicine & Biology*,

66(21), p.215018.

Zhang, Y., Shao, H.C., Pan, T. and Mengke, T., 2023. Dynamic cone-beam CT reconstruction using spatial and temporal implicit neural representation learning (STINR). *Physics in Medicine & Biology*, 68(4), p.045005.

Huang, Y., Thielemans, K., Price, G. and McClelland, J.R., 2024. Surrogate-driven respiratory motion model for projection-resolved motion estimation and motion compensated cone-beam CT reconstruction from unsorted projection data. *Physics in Medicine & Biology*, 69(2), p.025020.

It would be interesting to comment on whether you think the STO method could provide benefits for such approaches, and in the future to directly compare and/or integrate the STO approach with one of these methods (we would be very open to such discussions).

It is not clear to me if the STO method itself is being presented for the first time in this publication, or whether elements of it have been published before? If not, how does it relate to the similar methods previously published by your group (and others, if there are any). Please clarify.

Likewise, it is not clear if the adaptive reconstruction method is novel and being presented for the first time (although it sounds like it is).

How long does the adaptive reconstruction take to run in total (for both the STO600 and STO200 acquisitions)?

It would be interesting to see some examples of the MKB images. Did you consider using them directly (as an alternative to the conventional and adaptive recons)?

Once you have estimated the motion from the MKB images you calculate the reference image as the sum (or mean?) of the warped MKB images – did you consider taking the median instead (as is often done when creating mid-position images from 4DCT)? Or using the estimated motion to perform a motion compensated FDK reconstruction as in ref 20? This may result in images that are more similar to conventional 3D CBCTs (but hopefully without blurring and artefacts from motion) and may be more acceptable to clinical staff than the images from your adaptive reconstruction method.

line 70 of page 2, patient state -> motion state.

Page 9, line 373 "used to warp W the phase j"->"W is a function "

Page 9, Line 381, adaptive phase 6 -> adaptive phase r

Co-reviewed by

Jamie McClelland and Yuliang Huang

Version 1:

Reviewer comments:

Reviewer #1

(Remarks to the Author)

I believe the authors have responded well to both critiques. The Appendix provided important qualitative results regarding motion estimation, best and worst case, and all primary combinations of acquisition and reconstruction. I believe the authors satisfied almost every point in both of our reviews. I just have one remaining suggestion:

Following up on the other reviewer's comment regarding the motion captured by 4DCBCT, I think that although the motion may change during the treatment, it may still be worth tracking the tumor to ensure an agreement of tumor location between the conventional method and the STO within a reasonable margin. If they cannot be expected to be nearly similar, there would not be much value in 4DCBCT. We were asked to do this once for a paper, so we contoured the tumor in each image and tracked the tumor centroids. We calculated the distance between centroids in a given phase. This could also be done with the diaphragm boundary if visible. You don't have to do exactly this, but something to assess motion consistency seems valuable in a paper regarding the first clinical trial of this technique.

Here is that paper: Lauria, Michael, et al. "Motion compensated cone-beam CT reconstruction using an a priori motion model from CT simulation: a pilot study." *Physics in Medicine & Biology* 69.7 (2024): 075022.

Some other minor, mostly grammatical comments follow.

Minor comments:

1. Line 44: needs a comma after "reconstructed"
2. Line 237: is "spars" a typo, or something I'm unfamiliar with?
3. Line 315: "each ... images" -> each ... image
4. Line 359: Add "In" our implementation
5. Line 675: "Heart are avoided" -> heart is avoided

Reviewer #2

(Remarks to the Author)

We thank the authors for replying to and addressing our comments from our previous review. Most of our comments have been adequately addressed, however, there are still a couple of issues that we think should be addressed prior to publication.

We understand the authors comments about the problems with directly comparing motion in the conventional scan and STO scans from same session due to the time between the scans. However, doesn't the SSIM measure do exactly this, i.e. it compares the STO image to the conventional image? And presumably this is done between images from corresponding

phases, although this is not very clear from the text (furthermore, the text mentions 'windows' and 'regions' but does not explain what these are). If so the SSIM will be impacted by variations in the motion between the scans (as mentioned in the caption to fig 4 in the appendix), and so this metric is in some sense comparing the motion in the scans (although it is also impacted by other factors, such as the quality of the STO images), contradicting the new text on line 194. This issue should be clarified and discussed in the paper. Alternatively, there is an argument for removing the SSIM results, as it is difficult to draw a clear interpretation or conclusion from them, but we will leave this up to the authors.

We appreciate the authors adding the appendix as we requested, and agree it has improved the paper, but it still only includes examples from 3 (out of 30 x 2) different sessions, and one of these appears to be the same one used in the main paper. While this certainly demonstrates the robustness and consistency of the results better than just including a single example, including some more examples from other patients would help demonstrate this even better. It would also be good to include some examples with the worst TWI values to see how blurred the diaphragm/tumour are in these cases. We suggest removing fig 3 from the appendix (as this seems to just make the same point already made in figure 1), and then combining figs 4 and 5 and adding another 3-4 (or more!) examples, e.g. two with the worst TWI values and two with average / good numerical values.

Finally, the authors state that the journal does not allow animated figures, but the journal guidelines do mention videos and movies under supplementary material, so it seems these can be included as we previously suggested. We appreciate that the authors offered to make animations available on request, but readers are far more likely to view the movies if they are easily available as supplementary material rather than having to email to request them (in addition, as supplementary material they will be made available to us as reviewers to consider in our review). And we still think that including such movies will be of great benefit, allowing the reader to subjectively evaluate the results for themselves. It will also enable them to assess the plausibility of the motion seen in the scans, which is impossible to assess from just two images at peak inhale/exhale and is of particular interest for the adaptive recon scans where the observed motion will be impacted by the accuracy of the DIR results. If movies are added as supplementary material these could replace most/all of the figures in the appendix.

Co-reviewed by
Jamie McClelland and Yuliang Huang

Reviewer #3

(Remarks to the Author)
Review performed together with another reviewer - Jamie McClelland

Version 2:

Reviewer comments:

Reviewer #1

(Remarks to the Author)
I believe the revisions provided respond very well to comments by both reviewers. The motion consistency has been quantified and is indeed very consistent. All of my concerns have been addressed.

Reviewer #2

(Remarks to the Author)
Firstly, we would like to thank the authors for adding the supplementary material containing the various animations. In our opinion these really help to demonstrate how well the STO methods work, both in terms of the image quality and the ability to accurately capture the motion and are much easier to interpret than the quantitative numerical results. Looking at the animations, in particular the 3 'typical' cases, they show that the STO600 scans do indeed work well, and while they are a bit noisier and some of the structures are less clear (especially for case 1) the overall image quality and the motion they capture is similar to the conventional scan. However, in our opinion, the animations for the STO200 scans demonstrate the potential problems with this method, i.e. that the motion does not appear to be captured very well and the motion seen in the scans does not look very realistic/plausible, and that many of the structures appear 'blurred' compared to the conventional scan. This is particularly noticeable (and clinically relevant) for the tumour in cases 2 and 3, where the tumour in the STO scans appears blurrier and does not move as much as in the conventional (and STO600) scans (as can be seen in the coronal view for case 2 and the sagittal view for case 3). In our opinion it would be useful to briefly mention and discuss the issues seen in the animations of the STO200 method, otherwise it may give the reader the impression you had not noticed them or do not deem them important – but we also understand that this work is focusing on the quantitative evaluation of the methods in this paper, so will leave it to the authors if/how much they wish to discuss the subjective results seen in the animations. Note, lines 239-240 still say the animations are available on request, so should be updated.

We also like the fact that the authors have tried to include a quantitative evaluation of how similar the motion is in the two STO scans. We do not fully agree with the reasons for not also comparing the motion to the conventional scan – we appreciate that the time between the conventional scan and the STO scans is longer, but as the other reviewer said in their previous review, we would still expect the motion to be similar between the conventional and STO scans. However, if the authors feel strongly that the comparison to the conventional scan shouldn't be included that is up to them, but they should explain why they do not include the comparison as to many readers it may seem the more obvious comparison to make (as the conventional scan is considered the 'gold standard' that the STO scans are trying to match).

We also find it difficult to reconcile the numerical results from this new analysis and what we see subjectively in the animations. As mentioned above, it appears to us that the tumour motion is clearly underestimated in the STO200 animations compared to the STO600 animations, and by more than 1-2 voxels. We appreciate that the animations only show a few examples and we are subjectively assessing them, and that doing accurate quantitative analysis is very challenging, so its understandable that there may be some discrepancy. We are not really sure of the best way to address this (after all, the results are what they are) but at the moment the manuscript says that you have shown the STO200 scans accurately capture the motion (at least for the tumour) and then when we look at the animations this does not appear to be the case. Regarding the SSIM, we do not follow/agree with your claim that the SSIM will measure how well the large scale structure/motion is captured while being insensitive to small discrepancies in the structures/motion. The SSIM is sensitive to both small and large structures, and is just as sensitive to small misalignments of the structures as to large misalignments. The use of windows does not change this – it just provides local measures of the SSIM, indicating how well the structures in each window are aligned. As we previously commented, the SSIM will be impacted by both structures missing from the STO scans that are present in the conventional scan, and by structures being in different locations in the scans (either because the motion has changed between the scans, or because they have not been sufficiently well aligned). Indeed, I think there is a much stronger argument for not calculating SSIM between the conventional and STO scans due to the longer time between the scans (and hence differences in the locations of structures), than for not comparing the motion using your new analysis.

Which raises the question as to why you have very large SSIM values for both STO scans, and why the values are so similar for STO200 and STO600, when it can be seen in the images and even more clearly in the animations that some structures are missing from the STO scans or misaligned with the corresponding structures in the conventional images, and the appearance of the structures differs between the two STO scans – we initially put this down to SSIM not being very interpretable/meaningful (or useful imo) but looking more closely in appears there are a couple of mistakes in how you are calculating the SSIM. Firstly, when calculating c_1 and c_2 you should be using values of 0.01 and 0.03 (not 0.1 and 0.3 as in the manuscript). Secondly, although the original paper did set L to the full range of values allowed by the data type, this was because it was working with 8-bit video, where the full range of values (0-255) tended to be used in the images. When working with images that do not use the full range of values allowed by the data type (as is the case here) it is more appropriate to set L to be the max – min values seen in the images, i.e. the effective dynamic range of the images. This is what is used in this more recent publication focused on the use of SSIM in medical imaging:

https://aapm.onlinelibrary.wiley.com/doi/epdf/10.1002/mp.15514?saml_referrer

Another way to see this is a more appropriate way to set L is to consider saving all your images as 32-bit integers instead of 16-bit integers – this will have no impact on the images (other than doubling the size required to store them) so you would not expect/want it to impact the SSIM values, but if L is set according to the data type it will. The impact of these two mistakes is that both c_1 and c_2 will be much larger than they should be, and this will lead to values of SSIM that are very close to 1, regardless of what is in the images (as seen in your SSIM values).

Also, note, on line 501 you give the SSIM reference number as 12 but it should be 14.

We suggest the best way forward is to simply remove the SSIM analysis from the manuscript as we think it will add little even if it is calculated correctly for the reasons discussed above. If you do want to leave it in you need to ensure it is calculated correctly and remove or better justify the claim that it is insensitive to small structures/misalignments.

Co-reviewed by

Jamie McClelland and Yuliang Huang

Reviewer #3

(Remarks to the Author)

Review performed together with another reviewer - Jamie McClelland

Manuscript Title: Real-Time Spatiotemporal Optimization During Imaging

The authors would like to first thank the peer reviewers for their careful reviews. The manuscript has been altered in response to their input and we believe the revised manuscript will be much better received by the community as a result. In this letter we will go through each reviewer comment (in blue) and give our response (in black) with the relevant change to the manuscript included (in red).

The most substantial change to the manuscript has been an addition of an extensive appendix to display more of the acquired images for reader comparison. The appendix has also been attached to the end of this letter.

Reviewer #1 (Remarks to the Author):

Major claims: Scan time and scan dose are reduced using STO during 4D-CBCT while maintaining image quality and providing minimal impact on the clinical workflow. I believe all of these claims are backed, but if you claim to limit impact to the clinical workflow, I believe there should be some comments regarding the practical implications. Are there additional training, new equipment, additional setup or processing time, a need for therapists to interpret the successful or unsuccessful structure during acquisition, etc. It doesn't need to be too thorough, but there should be an acknowledgement of the impact on daily workflow and throughput other than reduced scan time and technical considerations.

We thank the reviewer for highlighting this oversight in the original manuscript. We have added the following to the manuscript:

Line 303 (addition in bold): **The conventional 4DCBCT scan in standard clinical practice acquires 1,320 2D x-ray projections over a $\theta_{arc} = 200^\circ$ arc and $\Delta_{arc} = 240$ seconds, a significant amount of time given that the treatment room may be booked for as little as 10 minutes per patient.**

Line 359: **Note that our implementation of STO we added equipment to the clinical environment namely the camera for surrogate signal monitoring, the control electronics and the control computer. However, a commercial implementation would not require separate control electronics and control computer, and the surrogate signal could be obtained by integrating the depth sensing camera into the treatment room or use current commercial surrogate monitoring solutions such as the Varian RPM. Beyond that there is no modification to the clinical environment or workflow due to implementing STO imaging in this context.**

Novelty: The approach is quite novel and impactful, so this paper describing the first results on patients in a clinical trial is very important, in my opinion. If other clinics implement this approach, they can achieve dose-efficient, and perhaps more importantly, time-efficient 4D-CBCTs with maintained image quality. This would be hugely beneficial.

Statistical Analysis: I think some key metrics are employed to characterize the important aspects of these imaging techniques: MAE to capture optimized angular distribution, SSIM to capture anatomical structures, CNR to capture image quality, and TIW to capture image sharpness. Summary statistics and paired t-tests are used to interpret. Based on the spread of data in Figure 6, I think the t-tests are appropriate.

Major comments:

1. This is up to the authors, but might you want to spend some time in the introduction describing the literature behind STO, and building up to the clinical trial?

The key literature leading up to the deployment of STO in the clinical trial has been referenced in the manuscript and we have added the following paragraph to the introduction to clarify how they relate to the presented work, line 58: *We present STO in this paper as a general framework for 4D imaging in which the data acquisition responds to a surrogate signal so that the reconstructed 4D image will have the best image quality relative to the scan time or total amount of acquired data. Sophisticated control in response to surrogate signals was implemented in [11] however in that case the focus was on isolating cardiac and respiratory motion in the acquired data so that the 3D reconstructed image had minimal motion blur. The phantom study presented in [12] acquires data in response to a respiratory surrogate signal based on a heuristic observation on the relationship between data structure and image quality with conventional 4D reconstruction. The simulation study presented in [30] developed reconstruction algorithms in response to structured data and demonstrated that certain 4D algorithms perform particularly well when data has certain structures.*

2. The first paragraph of the results would be greatly aided by a table. It would just make it faster and easier for readers to compare results across acquisitions/reconstructions.

Agreed. We have added the following:

Scan Characteristics

Scan Name	Reconstruction Method	Number of Projections	Mean Scan Time (s)	Data Structure Accuracy (°)
Conventional	Conventional	1,320	240	N/A
STO600	Conventional	600	242	0.71
STO200	Motion Compensated	200	91	1.75

Image Quality Quantification

Scan Name	SSIM	CNR	TIW-T (mm)	TIW-D (mm)
Conventional	1	7.5	7.8	7.7
STO600	0.988	5.9	10.2	9.4
STO200	0.989	12.4	5	3.5

Table 1: Summary of the acquired scans and reconstructed image quality.

3. There are two scan sessions acquired for many patients, resulting in 102 scans. However, there seems to be less than 102 data points in Figure 6. Were some removed? Maybe it is just hard to see.

Figure 6 contains all 102 data points however the jitter resulted in some points being almost overlaid. The data point transparency has been adjusted to help clarify where points overlap. Updated figure below:

Figure 6: Image quality metrics. Note that “better” performance by each metric is higher SSIM, higher CNR, lower TIW. SSIM is computed relative to the conventional image hence value of 1.

4. I think you could expand Figures 5 and 6 to include both acquisitions and reconstructions to better show off the impact of the adaptive reconstruction alone. At worst, you could expand Figure 5 to include these and leave Figure 6 alone. I fully understand the justification for highlighting STO600 conventional and STO200 adaptive, but still think Figure 5 is a good opportunity to further illustrate the direct impact of the reconstruction technique.

We have chosen to add a substantial appendix to the manuscript as both reviewers have expressed an interest in seeing more of the reconstructed images. We have expanded the caption of figure 5 to emphasize the change in image quality due to adaptive reconstruction (addition in bold):

Figure 5: Central coronal slice tomographs for conventional 4DCBCT and STO imaging at different combinations of scan time and reconstruction algorithm. **The observed reduction in streaking in the adaptive reconstruction is due to the STO600 adaptive reconstruction being computed from 600 projections and the STO200 adaptive reconstruction being computed from 200 projections, around which point acquiring further projections leads to minimal improvement in filtered backprojection image quality. The conventional reconstruction images from the conventional, STO600 and STO200 are computed from 132, 60 and 20 projections respectively i.e. just the projections acquired at that phase.**

Minor comments:

1. Author affiliation 3 is unassigned.

Fixed.

2. There are a few run-on sentences that could use commas or breaks, specifically in lines 23, 44, 323, 439, 557.

Fixed.

3. Lines 103-105: You already introduced the shorthand for STO600 and STO200, so you can shorten these.

Agreed, removed.

4. Line 109: First sentence sounds awkward.

Rewritten for clarity.

5. Line 336: Remove “just”

Fixed.

6. Lines 118-120: If the t-test was significant, instead of saying CNR was “comparable” for STO600, I’d maybe say “slightly reduced.”

Agreed, revised.

7. Line 309: (4) not really worded as a constraint in this list of constraints.

Agreed, revised.

8. Line 392: Capitalize “acquisition”?

Fixed.

9. Line 443: You could also cite <https://doi.org/10.1002/mp.12153> as I believe it is a very similar method.

Agreed. We added the following at the end of that paragraph:

Similar methods have been used for image quality quantification in motion compensated 4DCBCT [14] and 4DCT [36].

10. Great job on all figures – each one is presented well and greatly enhances understanding.

Thank you.

Reviewer #2 (Remarks to the Author):

This paper presents Spatiotemporal Optimisation (STO) Imaging, a method of actively controlling the acquisition of medical imaging data during free breathing such that the data

has a desired spatial-temporal structure. The paper presents this as a general approach but focusses on applying it to and evaluating it for 4DCBCT imaging. The paper also presents a complementary adaptive reconstruction algorithm that can be used together with the STO acquisition and provides improved images, especially when the number of projections is at its lowest (200). Finally, the paper also presents a clinical trial that performed STO acquisitions on 30 lung cancer patients and analyses the results.

Overall, the paper is well written and easy to follow, although I do find putting the methods section last, after the results and discussion, a confusing way of structuring a paper (but presume this is the format requested by the journal – if not I suggest changing it).

This is the format requested by the journal, we agree that as this method focuses on a new methodology it is somewhat confusing but we defer to the journal guides.

In general, the presented methods seem sensible (although I do have some questions and comments on them – see below), and the clinical trial has collected an impressive amount of data in order to evaluate them. However, I think the main weakness of the paper currently is the analysis of the results – currently this does not provide very compelling evidence that the method provides images that are suitable for clinical use, and I think it would benefit from improvements in this area before publication. Below I provide a number of specific comments and suggestions on how the paper could be improved – note, I appreciate that the paper already contains an impressive amount of content, with presenting the STO method, the adaptive reconstruction, and the clinical trial to evaluate these, so including all of my suggestions may be beyond the scope of the current paper.

We agree with your assessment that assessing the clinical utility of the images is a difficult and somewhat qualitative task, while the manuscript focuses on quantitative image analysis as we felt this would limit bias as and provide the type of summary information of interest to the journal audience. In order to allow readers to assess clinical utility, we have added an appendix showing a wide sample of the images collected in the trial.

One important aspect currently missing from the analysis is that of the motion seen in the 4DCBCT scans – the whole point of 4D (as opposed to 3D) scans is to see how the tumour and other anatomy moves, so a critical question is whether the motion appears the same in the STO scans as it does in the conventional scans? This is especially pertinent for the adaptive reconstructions, where errors in the registrations will impact the perceived motion.

We agree that the key clinical aspect of 4D imaging is representing not just the anatomy but the extent to which that anatomy moves. We have added an appendix to allow readers to assess how well anatomy and anatomy motion is captured in the acquired scans. A particularly challenging aspect in this study is that patient motion changes over the course of therapy, limiting the ability to compare motion observed in conventional and STO scans, so any analysis of motion needs to be restricted to during the particular scan acquisition. Because the reconstruction methods are all based on filtered backprojection, any errors in accounting for motion would manifest as a blurred tissue edge hence we use TIW as a proxy for motion representation accuracy. This challenge has been highlighted in the discussion section, with additions in bold, line 184:

Deformable image registration when used clinically often involves a manual verification step, suggesting a degree of mistrust[21]. In the case of adaptive reconstruction however, the final image is essentially a weighted average of 10 deformable registration results so a single poor registration would visually manifest as a blurred piece of anatomy [30], which is visibly tractable. The relatively good TIW values for STO200 images at the diaphragm and tumor boundaries would suggest that

representative motion is being captured, at least at these locations with the largest and most critical motion respectively.

The key clinical utility of 4D imaging is visualization of where the patient anatomy moves during the respiratory cycle, so the 4D image must represent motion accurately. An advantage of only using filtered backprojection based reconstruction methods in this study is that any error in accounting for motion of a particular piece of anatomy will manifest as a blurred edge [30], so TIW can be used as a proxy for motion accuracy. We chose not to compare the observed motion in scans within a particular fraction as previous work has demonstrated that patient motion can change by a factor of 1.5-1.6 over this time scale [37] hence the restriction to intrascan measures of motion accuracy. We show a wide range of images in the appendix of this manuscript so that readers can assess the accuracy to which each scan represents motion and general clinical utility.

I presume the SSIM is calculated for all phase images, so will be impacted by errors in estimating the motion, but it is also impacted by many other factors, and I think separate analysis of the ability of the STO scans to correctly estimate the motion is warranted. However, I appreciate that finding good ways to do this quantitatively is challenging (which leads to my next comment).

While I agree that it is necessary to include quantitative analysis as you have, I also think that qualitative analysis extremely valuable for assessing the quality of reconstructed images (and estimated motion). I would like to see many more examples of the reconstructed images from all methods, including the best and worse cases, as well as more typical examples – if there is not space to include these in the main paper they could be an appendix of supplementary material (although I think the main paper needs at least 2-3 examples to get any feel for how well the different methods have worked, e.g. based on the one example you have shown I question your assertion that the ST600-conv-recon is comparable to the conventional acquisition).

Agreed, ensuring that the motion is captured accurately is a key concern for 4D imaging in radiation therapy. We structured our quantitative analysis to try to reflect whether motion has been captured, but there is no standard method of doing this. With this concern in mind however we have added a substantial appendix to the manuscript as per your suggestion so that the readers can qualitatively assess for themselves whether they think the proposed method is accurately reflecting the underlying 4D anatomy.

In terms of quantitative measure of motion accuracy, this was the motivation for the STO600 acquisition scan with conventional reconstruction, and the use of a filtered backprojection based reconstruction for the STO200 scan. It would be difficult to assess from SSIM whether the tumor motion has been captured accurately, however the backprojections would produce blur at tissue interfaces if intrabrain motion or motion compensation was not being applied accurately, as explained in the manuscript, line 185:

In the case of adaptive reconstruction however, the final image is essentially a weighted average of 10 deformable registration results so a single poor registration would visually manifest as a blurred piece of anatomy [30], which is visibly tractable. The relatively good TIW values for STO200 images at the diaphragm and tumor boundaries would suggest that representative motion is being captured, at least at these locations with the largest and most critical motion respectively.

The key clinical utility of 4D imaging is visualization of where the patient anatomy moves during the respiratory cycle, so the 4D image must represent motion accurately. An advantage of only using filtered backprojection based reconstruction methods in this study is that any error in accounting for motion of a particular piece of anatomy will manifest as a blurred edge [30], so TIW can be used as a proxy for motion accuracy. We chose not to compare the observed motion in scans within a particular fraction as previous work has demonstrated that patient motion can change by a factor of 1.5-1.6 over

this time scale [37] hence the restriction to intrascan measures of motion accuracy. We show a wide range of images in the appendix of this manuscript so that readers can assess the accuracy to which each scan represents motion and general clinical utility. An emerging technique in 4DCBCT is the use of dynamic reconstruction methods that reconstruct the anatomy at each projection without assuming reproducible motion [40][41][42]. Use of these reconstruction methods with STO acquired data may provide clinicians with a more complete view of the range of patient motion, acquiring data that represents each patient breath individually during the scan acquisition. In this study we chose to restrict to the periodic 4D scan paradigm as that is more clinically familiar, and the low TIW values suggest that the assumption was valid for this patient cohort.

As per your suggestion for qualitative accuracy we have added an extensive appendix to the manuscript, reproduced at the end of this letter for your review. The appendix contains figures intended to demonstrate the impact of different reconstruction methods across the acquisitions, give example images demonstrating patient motion, and draw attention to the cases with the worst SSIM and CNR performance.

Likewise, movies of the results should be made available either as supplementary material (if allowed) or via a link to download them, as they will be very useful for assessing how well the STO methods can reproduce both the image quality and motion seen in the conventional images. The qualitative evaluation could be made more formal and semi-quantitative by getting clinical staff to rate the images, which could add valuable evidence as to whether the images are suitable for replacing conventional scans clinically. It would also be useful to link the quantitative and qualitative evaluations, e.g. state the quantitative metrics for the cases shown and/or show the images for the cases with the best/worst metrics.

We agree that as the study is concerned with 4D imaging we would ideally provide animated figures for the online version of the article or in supplementary figures, however this is not supported by the journal. However, we have clarified in the appendix that animations of the provided figures are available on request. As for the qualitative analysis, we have added the following to the discussion to clarify our position, line 217:

This work has been structured to demonstrate STO is performing in quantitative terms as much as possible e.g. by restricting to filtered backprojection type algorithms and analyzing tissue interface width, and by pursuing reductions in scan time and dose while quantitatively maintaining image quality so as to have minimal impact on the clinical decision making process. However, the key concern is whether the images are suitable for clinical use which necessarily involves a degree of qualitative assessment. Unfortunately, qualitative assessment by a panel of clinicians is a lengthy and expensive process and still subject to inter- and intra-observer variations. Another concern is that the qualitative difference in the images produced by each scan would complicate our ability to blind participants and introduce bias to the results. We therefore restrict this manuscript on the formalization and implementation of STO to a quantitative analysis as a necessary if not sufficient condition that the STO framework can maintain image quality relative to the amount of acquired data. Future work would involve qualitative investigation by clinicians, but the complexity and application specificity of such work would be more suitable for a discipline-specific journal. Readers can make their own qualitative assessment from images provided in the appendix, with animated versions available on reasonable request to the authors.

Related to the above, it would be interesting to include a brief discussion on exactly how the 4DCBCT images are used clinically, both now and how they may be used in the future (e.g. for auto-segmentation, for dose calculations and warping), as this will influence what is considered clinically acceptable image quality.

We have added the following on the recommended clinical use case for 4DCBCT in lung cancer radiation therapy, line 72:

In lung cancer radiation therapy European practice guidelines recommend 4DCBCT [38] particularly when treating in fewer treatment sessions, as it is critical to ensure the tumor is well contained in the

high dose region while avoiding radiosensitive organs as the patient breathes.

I think it would be interesting to further analyse the results for STO600-adapt-recon and STO200-conv-recon – while STO200-conv-recon is clearly worse than STO600-conv-recon visually, it would be informative to see how the metrics differed for all 4 images, and would help to interpret the metric results for the STO600-conv-recon and STO200-adapt-recon that have been provided. It would also help to assess the separate effects of varying the number of projections and the reconstruction method on the image quality and the estimated motion. Furthermore, from the one example shown it looks like STO600-adapt-recon may even give better results than the conventional imaging (which if it is correcting for the motion well it should do) – although this is very challenging to show when you consider the conventional images as the gold standard you are comparing to.

Based on the reasons outlined below we have chosen to restrict the analysis to STO600-conv and STO200-adapt to simplify the manuscript and restrict to just 2 comparisons against the current clinical standard – similar speed, half dose, regular reconstruction and faster, very low dose, motion compensated reconstruction. We have added the following expanded explanation to the results, line 110:

Central slice coronal tomographs are shown in figure 5 for a conventional 4DCBCT (conventional acquisition, conventional reconstruction) as well as the STO600 and STO200 acquisitions with conventional and adaptive reconstruction. The STO600 acquisition conventional reconstruction scan has comparable image quality to conventional acquisition conventional reconstruction scan while reducing scan dose just by altering the acquisition. The STO200 acquisition adaptive reconstruction scan allows large reductions in scan time and dose while approximately maintaining image quality thanks to the adaptive reconstruction method working well with the STO data structure. **The STO200 acquisition conventional reconstruction scans had clearly unacceptable image quality, and the STO600 acquisition adaptive reconstruction scan does not demonstrate a sufficient improvement in image quality over the STO200 adaptive reconstruction scan to justify the additional scan dose and time while still having the clinical concerns around use of motion compensated reconstruction. We have therefore omitted analysis of these scans, however more extensive investigations of the relationship between acquisition and reconstruction can be found in [30] [39].**

The STO method is well suited to addressing a specific problem with conventional 4DCBCT, that the projections are unevenly sampled (relative to phase), and hence many of them provide limited information. However, this is not the only potential problem/limitation with 4DCBCT. Another is that it, like many 4D imaging methods, assumes reproducible motion from one breath to another, and that the motion is strongly related to the surrogate signal used to measure the respiratory phase – and these assumptions do not always hold. While this study does not aim to address these issues, it does provide an interesting opportunity to study their potential impact on your proposed methods, especially as you noted that there were some individuals/scans that experienced periods of very irregular breathing. It would be good to discuss these cases in a bit more detail and in relation to the issue of irregular motion, and to show the images for these cases. Related to this, there have recently been some published methods that attempt to model the variable motion throughout the acquisition and do not rely on the assumption of reproducible motion, e.g.

Jailin, C., Roux, S., Sarrut, D. and Rit, S., 2021. Projection-based dynamic tomography. *Physics in Medicine & Biology*, 66(21), p.215018.

Zhang, Y., Shao, H.C., Pan, T. and Mengke, T., 2023. Dynamic cone-beam CT reconstruction using spatial and temporal implicit neural representation learning (STINR). *Physics in Medicine & Biology*, 68(4), p.045005.

Huang, Y., Thielemans, K., Price, G. and McClelland, J.R., 2024. Surrogate-driven respiratory motion model for projection-resolved motion estimation and motion compensated cone-beam CT reconstruction from unsorted projection data. *Physics in Medicine & Biology*, 69(2), p.025020.

It would be interesting to comment on whether you think the STO method could provide benefits for such approaches, and in the future to directly compare and/or integrate the STO approach with one of these methods (we would be very open to such discussions).

We have restricted our analysis to data acquired within each scan due to the interscan variation in patient motion. Because the reconstruction methods are based on filtered backprojection, any error in accounting for intrascan motion due to e.g. intrabrain variability would be represented as blur at the tissue boundary, so we use TIW as a proxy for intrascan variation. We have expanded the explanation of this effect, line 190:

The key clinical utility of 4D imaging is visualization of where the patient anatomy moves during the respiratory cycle, so the 4D image must represent motion accurately. An advantage of only using filtered backprojection based reconstruction methods in this study is that any error in accounting for motion of a particular piece of anatomy will manifest as a blurred edge [30], so TIW can be used as a proxy for motion accuracy. We chose not to compare the observed motion in scans within a particular fraction as previous work has demonstrated that patient motion can change by a factor of 1.5-1.6 over this time scale [37] hence the restriction to intrascan measures of motion accuracy. We show a wide range of images in the appendix of this manuscript so that readers can assess the accuracy to which each scan represents motion and general clinical utility. An emerging technique in 4DCBCT is the use of dynamic reconstruction methods that reconstruct the anatomy at each projection without assuming reproducible motion [40][41][42]. Use of these reconstruction methods with STO acquired data may provide clinicians with a more complete view of the range of patient motion, acquiring data that represents each patient breath individually during the scan acquisition. In this study we chose to restrict to the periodic 4D scan paradigm as that is more clinically familiar, and the low TIW values suggest that the assumption was valid for this patient cohort.

It is not clear to me if the STO method itself is being presented for the first time in this publication, or whether elements of it have been published before? If not, how does it relate to the similar methods previously published by your group (and others, if there are any). Please clarify. Likewise, it is not clear if the adaptive reconstruction method is novel and being presented for the first time (although it sounds like it is).

This is the first study to present the abstracted framework of STO and the first to present full results of the first STO clinical trial. We have published earlier specific examples of STO, some preliminary data from the trial, and the adaptive reconstruction method. The existing literature has been cited throughout the manuscript. The key literature leading up to the deployment of STO in the clinical trial has been referenced in the manuscript and we have added the following paragraph to the introduction to clarify how they relate to the presented work:

Line 58: We present STO in this paper as a general framework for 4D imaging in which the data acquisition responds to a surrogate signal so that the reconstructed 4D image will have the best image quality relative to the scan time or total amount of acquired data. Sophisticated control in response to surrogate signals was implemented in [11] however in that case the focus was on isolating cardiac and respiratory motion in the acquired data so that the 3D reconstructed image had minimal motion blur. The phantom study presented in [12] acquires data in response to a respiratory signal based on a heuristic observation on the relationship between data structure and image quality with conventional 4D reconstruction. The simulation study presented in [30] developed reconstruction algorithms in response to structured data and demonstrated that certain 4D algorithms perform particularly well when data has certain structures.

How long does the adaptive reconstruction take to run in total (for both the STO600 and STO200 acquisitions)?

The following has been added, line 436:

In our implementation 4DFDK reconstruction took approximately 1 minute and MKB reconstruction took approximately 2 minutes. The 9 registrations for adaptive reconstruction added approximately 18 minutes so in total the adaptive reconstruction algorithm required 20 minutes however with sufficient computing the registrations could be performed in parallel to reduce computation time to 4-5 minutes using the current research implementation of our software. We believe that refining the software to avoid unnecessary read/write overheads, making use of hardware acceleration and particularly the integration of AI registration methods for inference [43] the computation time could be reduced to under half a minute.

It would be interesting to see some examples of the MKB images. Did you consider using them directly (as an alternative to the conventional and adaptive recons)?

We have included example MKB images in the appendix. Our impression was that STO200 acquisition MKB reconstruction images reduced image quality relative to the current standard, so arguing for their clinical use would be difficult.

Once you have estimated the motion from the MKB images you calculate the reference image as the sum (or mean?) of the warped MKB images – did you consider taking the median instead (as is often done when creating mid-position images from 4DCT)? Or using the estimated motion to perform a motion compensated FDK reconstruction as in ref 20? This may result in images that are more similar to conventional 3D CBCTs (but hopefully without blurring and artefacts from motion) and may be more acceptable to clinical staff than the images from your adaptive reconstruction method.

We chose to form images from the mean as this ensures all acquired information is still represented in the final image, in this case manifesting similarly to motion blur. We believe this is a conservative approach that is suitable when pursuing clinical translation, with the concerns and rationale give in line 179:

The STO200 images are visibly better than conventional images, and the image quality metrics support this claim. This suggests that the presented STO method is sufficiently robust even for the relatively large spatiotemporal variation in the data observed in 2 STO scans. While the STO200 scan uses mean 63% less scan time and 85% less scan dose while delivering quantifiably improved image quality, the use of deformable image registration to perform motion compensated reconstruction[20] may hinder clinical uptake. Deformable image registration when used clinically often involves a manual verification step, suggesting a degree of mistrust[21]. In the case of adaptive reconstruction however, the final image is essentially a weighted average of 10 deformable registration results so a single poor registration would visually manifest as a blurred piece of anatomy [30], which is visibly tractable. The relatively good TIW values for STO200 images at the diaphragm and tumor boundaries would suggest that representative motion is being captured, at least at these locations with the largest and most critical motion respectively.

line 70 of page 2, patient state -> motion state.

Fixed.

Page 9, line 373 “used to warp W the phase j ”->“ W is a function ”

Fixed.

Page 9, Line 381, adaptive phase 6 -> adaptive phase r

Fixed.

Appendix

Evaluating the clinical utility of images depends on a wide range of factors that are often user dependent, especially for scans of real patients where no ground truth is available. In this manuscript we implemented STO imaging in the specific application of 4DCBCT imaging for radiation therapy, where the typical use case is to verify that the tumor remains within the treatment beam across the respiratory cycle while highly radiosensitive tissues e.g. the heart are avoided.

We have chosen to include this appendix so that readers can make their own qualitative assessment of the images. Note that in this trial the patients receive the conventional 4DCBCT scan, are aligned and treated as in typical clinical practice, then the STO600 scan is acquired, and then the STO200 scan is acquired. We therefore expect to see a rigid translation between the conventional and adaptive images. As explained in the text of the manuscript, it is also common for the patients respiratory motion to qualitatively change over this time, with up to 60% change in the motion amplitude [37]. The considerations should be kept in mind when evaluating the impact of STO acquisition and adaptive reconstruction on the observed motion in 4D images.

Another consideration in assessing CBCT images is that the distribution of attenuation values changes significantly depending on the acquisition type and reconstruction algorithm. A consequence is CBCT images are usually manually windowed to increase contrast of the relevant tissues. The images presented have been manually windowed to give the highest qualitative similarity.

Appendix Figure 1: Comparison of acquisitions and reconstructions for an example patient. Note that all the conventional acquisition images appear to have similar image quality, likely due to filtered backprojection reconstruction quality saturating at this high number of projections. The impact of different reconstruction algorithms is most pronounced in the STO200 scan, where conventional FDK reconstruction image quality is noticeably low.

Appendix Figure 2: Comparison of conventional acquisition conventional reconstruction, STO600 acquisition conventional reconstruction, and STO200 acquisition adaptive reconstruction scans at peak inhale and peak exhale for an example patient.

Appendix Figure 3: Comparison of conventional acquisition with conventional (FDK), McKinnon-Bates (MKB) and Adaptive reconstruction at peak exhale and inhale for an example patient.

Appendix Figure 4: Patient with lowest SSIM between conventional and STO200 images at peak inhale. It appears this patient began taking shallower breaths over the course of the treatment.

Appendix Figure 5: Patient with worst STO200 CNR relative to conventional scan. Observe that all scans have the same FOV so this is a relatively large patient, so we expect in general a lower CNR for a fixed kVp and mAs per projection as in our setup. The relative poor CNR of the STO200 image relative to the conventional image may be due to simple variation in CNR at low values, or the projection weighting during the filtered backprojection process not calculating accurately for the low available signal across few projections in the STO200 scan.

Hello,

Thank you to the reviewers and editors for their thoughtful comments. We have revised the manuscript in response. In this response letter we have highlighted reviewer comments in red and manuscript changes in blue.

There are two substantial changes to the manuscript in this submission. The first is the application of a metric based on the work of Lauria, Michael, et al to analyze whether the motion compensated images are placing the tumor in the correct location. The second is the addition of a supplementary materials file containing animations of the conventional, STO600 and STO200 scans. These additions will help readers verify that STO maintains image quality, while enabling 85% reductions in scan dose and 65% reductions in scan time.

Reviewer #1 (Remarks to the Author):

I believe the authors have responded well to both critiques. The Appendix provided important qualitative results regarding motion estimation, best and worst case, and all primary combinations of acquisition and reconstruction. I believe the authors satisfied almost every point in both of our reviews. I just have one remaining suggestion:

Following up on the other reviewer's comment regarding the motion captured by 4DCBCT, I think that although the motion may change during the treatment, it may still be worth tracking the tumor to ensure an agreement of tumor location between the conventional method and the STO within a reasonable margin. If they cannot be expected to be nearly similar, there would not be much value in 4DCBCT. We were asked to do this once for a paper, so we contoured the tumor in each image and tracked the tumor centroids. We calculated the distance between centroids in a given phase. This could also be done with the diaphragm boundary if visible. You don't have to do exactly this, but something to assess motion consistency seems valuable in a paper regarding the first clinical trial of this technique. Here is that paper: Lauria, Michael, et al. "Motion compensated cone-beam CT reconstruction using an a priori motion model from CT simulation: a pilot study." *Physics in Medicine & Biology* 69.7 (2024): 075022.

Thank you for the suggestion. The proposed method seems useful as a way of measuring motion consistency when motion compensated reconstruction methods are used, so we have decided to implement a similar approach, using deformable image registration rather than manual contouring to reduce user bias. We have adjusted the manuscript to reflect this:

Line 516: A concern in motion compensated imaging is whether the motion model is robust, particularly in the tumor region. A method of testing motion model robustness proposed in [44] is to contour the tumor in images reconstructed with and without motion compensation and compare the centroid location. We implemented a more automated approach in which the motion compensated STO200 images are deformably registered to the STO600 images and the DVF values in the tumor are taken as a volumetric displacement as described in [45]. We compute a mean displacement in each axis as well as a total distance and compute standard deviations.

Line 144: We measured motion model robustness [45] by deformably registering motion compensated STO200 images to the non-motion compensated STO600 images and extracting the motion in the tumor region as per [46]. We observed mean displacements of 0.28mm, -0.0041mm and 0.11mm with standard deviation 0.70mm, 0.75mm and 0.68mm in the superior/inferior, posterior/anterior and left/right axes respectively. The total

displacements had mean of 1.2mm and standard deviation 0.47mm. These results suggest strong consistency in tumor positioning [45] well within the anatomically expected amount of variation over the acquisition time for both scans [37].

Line 197: The key clinical utility of 4D imaging is visualization of where the patient anatomy moves during the respiratory cycle, so the 4D image must represent motion accurately. An advantage of only using filtered backprojection based reconstruction methods in this study is that any error in accounting for motion of a particular piece of anatomy will manifest as a blurred edge [30], so TIW can be used as a proxy for motion accuracy. There are concerns about motion model robustness when motion compensated reconstruction is used so we perform a similar analysis to [45] to confirm that the tumour location is consistent between the non-motion compensated STO600 scan and motion compensated STO200 scan acquired immediately after. We found that the tumour was within 1-2 voxels for each scan, confirming the motion model robustness considering the expected anatomic variation over the acquisition of both scans [37]. We show a wide range of images in the appendix and supplementary materials of this manuscript so that readers can assess the accuracy to which each scan represents motion and general clinical utility. An emerging technique in 4DCBCT is the use of dynamic reconstruction methods that reconstruct the anatomy at each projection without assuming reproducible motion [40][41][42]. Use of these reconstruction methods with STO acquired data may provide clinicians with a more complete view of the range of patient motion, acquiring data that represents each patient breath individually during the scan acquisition. In this study we chose to restrict to the periodic 4D scan paradigm as that is more clinically familiar, and the low TIW values suggest that the assumption was valid for this patient cohort.

Some other minor, mostly grammatical comments follow.

Minor comments:

1. Line 44: needs a comma after “reconstructed”
2. Line 237: is “spars” a typo, or something I’m unfamiliar with?
3. Line 315: “each ... images” -> each ... image
4. Line 359: Add “In” our implementation
5. Line 675: “Heart are avoided” -> heart is avoided

Thank you, fixed.

Reviewer #2 (Remarks to the Author):

We thank the authors for replying to and addressing our comments from our previous review. Most of our comments have been adequately addressed, however, there are still a couple of issues that we think should be addressed prior to publication.

We understand the authors comments about the problems with directly comparing motion in the conventional scan and STO scans from same session due to the time between the scans. However, doesn't the SSIM measure do exactly this, i.e. it compares the STO image to the conventional image? And presumably this is done between images from corresponding phases, although this is not very clear from the text (furthermore, the text mentions ‘windows’ and ‘regions’ but does not explain what these are). If so the SSIM will be impacted by variations in the motion between the scans (as mentioned in the caption to fig 4 in the appendix), and so this metric is in some sense comparing the motion in the scans (although it is also impacted by other factors, such as the quality of the STO images), contradicting the new text on line 194. This issue should be clarified and discussed in the

paper. Alternatively, there is an argument for removing the SSIM results, as it is difficult to draw a clear interpretation or conclusion from them, but we will leave this up to the authors.

Thank you for raising this issue. The purpose of the SSIM comparisons is to ensure that larger scale motion is captured in the images without focusing on specific small structures that are known to move differently over time. We have revised the section on SSIM to reflect this distinction:

Line 500: To quantify how well large scale structures in the images are captured, we compute the structural similarity index measure (SSIM)[12] as $SSIM =$

$\frac{1}{M} \sum_M \frac{(2\mu_{PCI}\mu_{conv}+c_1)(2\sigma_{PCI,conv}+c_2)}{(\mu_{PCI}^2+\mu_{conv}^2+c_1)(\sigma_{PCI}^2+\sigma_{conv}^2+c_2)}$ over M windows inside the region, μ_{PCI} and μ_{conv} are voxel value means of the STO and conventional image respectively in that window, σ_{PCI} , σ_{conv} and $\sigma_{PCI,conv}$ are standard deviation and joint deviation, $c_1 = (0.1L)^2$ and $c_2 = (0.3L)^2$ are weighting factors with L the dynamic range in this case $2^{16} - 1$ as reconstructions computed to 16 bit precision. Note that the use of windows in computing SSIM makes the metric useful for observing large sections of anatomy are moving in roughly the same way between scans but is insensitive to small motion discrepancies.

The importance however of capturing tumor motion in 4D images remains, as highlighted by reviewer 1. Fortunately, they proposed a method from a recent publication that seeks to quantify if tumor motion is consistent between motion compensated and non-motion compensated scans. We have implemented a similar approach and found the motion is consistent:

Line 516: A concern in motion compensated imaging is whether the motion model is robust, particularly in the tumor region. A method of testing motion model robustness proposed in [44] is to contour the tumor in images reconstructed with and without motion compensation and compare the centroid location. We implemented a more automated approach in which the motion compensated STO200 images are deformably registered to the STO600 images and the DVF values in the tumor are taken as a volumetric displacement as described in [45]. We compute a mean displacement in each axis as well as a total distance and compute standard deviations.

Line 144: We measured motion model robustness [45] by deformably registering motion compensated STO200 images to the non-motion compensated STO600 images and extracting the motion in the tumor region as per [46]. We observed mean displacements of 0.28mm, -0.0041mm and 0.11mm with standard deviation 0.70mm, 0.75mm and 0.68mm in the superior/inferior, posterior/anterior and left/right axes respectively. The total displacements had mean of 1.2mm and standard deviation 0.47mm. These results suggest strong consistency in tumor positioning [45] well within the anatomically expected amount of variation over the acquisition time for both scans [37].

Line 197: The key clinical utility of 4D imaging is visualization of where the patient anatomy moves during the respiratory cycle, so the 4D image must represent motion accurately. An advantage of only using filtered backprojection based reconstruction methods in this study is that any error in accounting for motion of a particular piece of anatomy will manifest as a blurred edge [30], so TIW can be used as a proxy for motion accuracy. There are concerns about motion model robustness when motion compensated reconstruction is used so we perform a similar analysis to [45] to confirm that the tumour location is consistent between the non-motion compensated STO600 scan and motion compensated STO200 scan acquired immediately after. We found that the tumour was within 1-2 voxels for each scan, confirming the motion model robustness considering the expected anatomic variation over

the acquisition of both scans [37]. We show a wide range of images in the appendix and supplementary materials of this manuscript so that readers can assess the accuracy to which each scan represents motion and general clinical utility. An emerging technique in 4DCBCT is the use of dynamic reconstruction methods that reconstruct the anatomy at each projection without assuming reproducible motion [40][41][42]. Use of these reconstruction methods with STO acquired data may provide clinicians with a more complete view of the range of patient motion, acquiring data that represents each patient breath individually during the scan acquisition. In this study we chose to restrict to the periodic 4D scan paradigm as that is more clinically familiar, and the low TIW values suggest that the assumption was valid for this patient cohort.

We appreciate the authors adding the appendix as we requested, and agree it has improved the paper, but it still only includes examples from 3 (out of 30 x 2) different sessions, and one of these appears to be the same one used in the main paper. While this certainly demonstrates the robustness and consistency of the results better than just including a single example, including some more examples from other patients would help demonstrate this even better. It would also be good to include some examples with the worst TWI values to see how blurred the diaphragm/tumour are in these cases. We suggest removing fig 3 from the appendix (as this seems to just make the same point already made in figure 1), and then combining figs 4 and 5 and adding another 3-4 (or more!) examples, e.g. two with the worst TWI values and two with average / good numerical values.

Finally, the authors state that the journal does not allow animated figures, but the journal guidelines do mention videos and movies under supplementary material, so it seems these can be included as we previously suggested. We appreciate that the authors offered to make animations available on request, but readers are far more likely to view the movies if they are easily available as supplementary material rather than having to email to request them (in addition, as supplementary material they will be made available to us as reviewers to consider in our review). And we still think that including such movies will be of great benefit, allowing the reader to subjectively evaluate the results for themselves. It will also enable them to assess the plausibility of the motion seen in the scans, which is impossible to assess from just two images at peak inhale/exhale and is of particular interest for the adaptive recon scans where the observed motion will be impacted by the accuracy of the DIR results. If movies are added as supplementary material these could replace most/all of the figures in the appendix.

Co-reviewed by
Jamie McClelland and Yuliang Huang

Thank you for the suggestion. We have decided to upload the Conventional, STO600 and STO200 animated images as supplementary material. We provide 6 cases as representative examples of the data set and include the worst performing cases as quantified by CNR and SSIM. We limit to 5 cases to keep under the 50MB supplementary material file limit. We have chosen to keep the appendix as is, as some readers may prefer still images.

Hello,

Thank you to the reviewers and editors for their thoughtful comments. We have revised the manuscript in response. In this response letter we have highlighted reviewer comments in red and manuscript changes in blue.

The top level changes to the manuscript is the requested removal of the SSIM image quality metric, and to further expand on potential issues with one of the scan types. We have softened the language around this scan type to reflect these potential issues. We also invite the reader at several points in the manuscript to view animations provided as supplementary material as we believe this is the best way to represent the results and allow the readers to objectively assess the novel scans presented in the work.

Reviewer #1 (Remarks to the Author):

I believe the revisions provided respond very well to comments by both reviewers. The motion consistency has been quantified and is indeed very consistent. All of my concerns have been addressed.

Thank you for your comments. We believe the motion consistency method has improved the quality of the analysis and are glad we have addressed your concerns.

Reviewer #2 (Remarks to the Author):

Firstly, we would like to thank the authors for adding the supplementary material containing the various animations. In our opinion these really help to demonstrate how well the STO methods work, both in terms of the image quality and the ability to accurately capture the motion and are much easier to interpret than the quantitative numerical results.

Thank you for encouraging us to provide this supplementary material, we are glad it has addressed some of your concerns and believe readers will feel similarly. For this reason, we start the image quality segment of our results section with the following to direct the reader's attention:

Line 110: The supplementary materials to this article includes central slice tomographs for 5 selected cases. We present and discuss static images in the manuscript, but we encourage the reader to view the supplementary material to make their own qualitative assessment of the images.

Looking at the animations, in particular the 3 'typical' cases, they show that the STO600 scans do indeed work well, and while they are a bit noisier and some of the structures are less clear (especially for case 1) the overall image quality and the motion they capture is similar to the conventional scan. However, in our opinion, the animations for the STO200 scans demonstrate the potential problems with this method, i.e. that the motion does not appear to be captured very well and the motion seen in the scans does not look very realistic/plausible, and that many of the structures appear 'blurred' compared to the conventional scan. This is particularly noticeable (and clinically relevant) for the tumour in cases 2 and 3, where the tumour in the STO scans appears blurrier and does not move as much as in the conventional (and STO600) scans (as can be seen in the coronal view for case 2 and the sagittal view for case 3). In our opinion it would be useful to briefly mention and discuss the issues seen in the animations of the STO200 method, otherwise it may give the reader the impression you had not noticed them or do not deem them important – but we

also understand that this work is focusing on the quantitative evaluation of the methods in this paper, so will leave it to the authors if/how much they wish to discuss the subjective results seen in the animations.

We agree that there are issues with the motion compensation in some of the STO200 scans, and these issues are best conveyed by reference to the animations in the supplementary material.

We have chosen to add the following paragraph to the start of the image quality section of the discussion, as we expect the animations in the supplementary material will most likely be the main aspect of the article that readers will want to engage with.

Line 178: Throughout this discussion we will refer not only to the quantitative metrics and figures provided within the manuscript, but also to features visible in the animations provided as supplementary material to this manuscript. Given the focus on 4D images in this study we encourage readers to view these supplementary materials and make their own qualitative assessment particularly with regards to the use of motion compensated reconstruction for the STO200 scan.

We added the following paragraph later in the discussion to highlight key observations we make from the images and provide context for how such images are used clinically.

Line 205: As we are considering a 4D modality we direct the readers to review the supplementary materials to this manuscript which contains central slice tomographs of some representative patient scans. As observed in the static images the conventional and STO600 images are fairly similar albeit with some additional noise in the STO600 images likely due to acquiring around half as many projections. As in the static images we observe that the STO200 images have some widespread blurring likely due to imprecise motion compensation, however additional artefacts become noticeable in the animations. Most immediately apparent is a general unrealistic flexing in the images particularly near the periphery. This is most likely due to the image registration in this region aligning streaks to streaks rather than meaningfully capturing moving anatomy. In practice the images presented to physicians are masked so that only the part of the image well contained in the geometric field of view is visible, so these edge artefacts would not be visible, but we include them here as they may be considered of interest to the research community. We observe in the case 2 STO200 scan that the tumor moves erratically near peak exhale as registration of this phase may be inaccurate, and this is reflected by the additional blurring of the tumor edge in this case. More accurate registration or a synthetic middle phase as in [19] may reduce these effects. We also observe in case 3 that the STO200 tumor has little motion relative to the conventional scan image, however we similarly observe reduced motion in the STO600 scan that does not use motion compensation, suggesting the possibility that this patient was gradually reducing their breathing amplitude.

Note, lines 239-240 still say the animations are available on request, so should be updated. Agreed, fixed.

We also like the fact that the authors have tried to include a quantitative evaluation of how similar the motion is in the two STO scans. We do not fully agree with the reasons for not also comparing the motion to the conventional scan – we appreciate that the time between the conventional scan and the STO scans is longer, but as the other reviewer said in their previous review, we would still expect the motion to be similar between the conventional and STO scans. However, if the authors feel strongly that the comparison to the conventional scan shouldn't be included that is up to them, but they should explain why they do not

include the comparison as to many readers it may seem the more obvious comparison to make (as the conventional scan is considered the 'gold standard' that the STO scans are trying to match).

We appreciate that a comparison to the gold standard conventional scan would be ideal however we observe considerable discrepancy in patient motion in the conventional scans acquired shortly after the patient lays on the couch, and the STO scans acquired ~20 minutes later. This can be seen in the supplementary materials for example case 3 in which the tumor exhibits substantial motion during the conventional scan, and almost imperceptible motion in the STO600 and STO200 scans. We have made the following addition to the discussion section to emphasize why the decision to compare STO600 and STO200 was made:

Line 228: The decision was made to compare STO600 and STO200 scans rather than back to the conventional as we do not expect the altered acquisition mode to impact patient motion but we do know patient motion changes over the ~20 minutes from conventional 4DCBCT acquisition through treatment delivery to STO acquisition [37].

We also find it difficult to reconcile the numerical results from this new analysis and what we see subjectively in the animations. As mentioned above, it appears to us that the tumour motion is clearly underestimated in the STO200 animations compared to the STO600 animations, and by more than 1-2 voxels. We appreciate that the animations only show a few examples and we are subjectively assessing them, and that doing accurate quantitative analysis is very challenging, so its understandable that there may be some discrepancy. We are not really sure of the best way to address this (after all, the results are what they are) but at the moment the manuscript says that you have shown the STO200 scans accurately capture the motion (at least for the tumour) and then when we look at the animations this does not appear to be the case.

Given that you have these concerns, we have chosen to soften the language throughout the manuscript to make clear that the STO200 scans do appear to have differences so may be suitable for some but not all clinics and patients. We maintain confidence in our quantitative analysis that the tumor centroid is being represented to within 1-2mm as verified by the motion model robustness metric and that the tumor boundary is being accurately represented as verified by the TIW but we do accept that a viewer presented with some of the images may interpret the tumor boundaries differently. It is for this reason that we choose to emphasize throughout the manuscript that the reader should view the animations provided in supplementary materials and make their own judgement if they think STO and in particular the STO200 scan is a sufficiently accurate technology that they would like implemented in their own clinic. We have softened language around the effectiveness of STO200 in the first discussion paragraph of image quality concerning STO200.

Line 191: The STO200 images initially appear qualitatively clearer than conventional images, and some of the image quality metrics support this claim. This suggests that the presented STO method may be sufficiently robust even for the relatively large spatiotemporal variation in the data observed in 2 STO scans. While the STO200 scan uses mean 63% less scan time and 85% less scan dose while delivering quantifiably improved image quality, the use of deformable image registration to perform motion compensated reconstruction[20] may hinder clinical uptake. Deformable image registration when used clinically often involves a manual verification step, suggesting a degree of mistrust[21] and does have the potential to introduce artefacts in the final images. We invite the viewer to view the images in this

manuscript and the animations in the supplementary material to make their own judgement if the motion compensation is sufficiently reliable for the clinical use case.

Regarding the SSIM, we do not follow/agree with your claim that the SSIM will measure how well the large scale structure/motion is captured while being insensitive to small discrepancies in the structures/motion. The SSIM is sensitive to both small and large structures, and is just as sensitive to small misalignments of the structures as to large misalignments. The use of windows does not change this – it just provides local measures of the SSIM, indicating how well the structures in each window are aligned. As we previously commented, the SSIM will be impacted by both structures missing from the STO scans that are present in the conventional scan, and by structures being in different locations in the scans (either because the motion has changed between the scans, or because they have not been sufficiently well aligned). Indeed, I think there is a much stronger argument for not calculating SSIM between the conventional and STO scans due to the longer time between the scans (and hence differences in the locations of structures), than for not comparing the motion using your new analysis.

Which raises the question as to why you have very large SSIM values for both STO scans, and why the values are so similar for STO200 and STO600, when it can be seen in the images and even more clearly in the animations that some structures are missing from the STO scans or misaligned with the corresponding structures in the conventional images, and the appearance of the structures differs between the two STO scans – we initially put this down to SSIM not being very interpretable/meaningful (or useful imo) but looking more closely it appears there are a couple of mistakes in how you are calculating the SSIM. Firstly, when calculating c_1 and c_2 you should be using values of 0.01 and 0.03 (not 0.1 and 0.3 as in the manuscript). Secondly, although the original paper did set L to the full range of values allowed by the data type, this was because it was working with 8-bit video, where the full range of values (0-255) tended to be used in the images. When working with images that do not use the full range of values allowed by the data type (as is the case here) it is more appropriate to set L to be the max – min values seen in the images, i.e. the effective dynamic range of the images. This is what is used in this more recent publication focused on the use of SSIM in medical imaging:

https://aapm.onlinelibrary.wiley.com/doi/epdf/10.1002/mp.15514?saml_referrer

Another way to see this is a more appropriate way to set L is to consider saving all your images as 32-bit integers instead of 16-bit integers – this will have no impact on the images (other than doubling the size required to store them) so you would not expect/want it to impact the SSIM values, but if L is set according to the data type it will. The impact of these two mistakes is that both c_1 and c_2 will be much larger than they should be, and this will lead to values of SSIM that are very close to 1, regardless of what is in the images (as seen in your SSIM values).

Also, note, on line 501 you give the SSIM reference number as 12 but it should be 14.

We suggest the best way forward is to simply remove the SSIM analysis from the manuscript as we think it will add little even if it is calculated correctly for the reasons discussed above.

If you do want to leave it in you need to ensure it is calculated correctly and remove or better justify the claim that it is insensitive to small structures/misalignments.

We agree that SSIM has several failings as an image metric, and we have generally included it as an additional data point that some readers may or may not consider of interest. However, as you point out there are several constants that can be chosen differently based on how the images are represented and weighted that can make comparisons between papers difficult. With these considerations in mind we have chosen to take your recommendation and remove SSIM from the manuscript, with the remaining set of metrics

being more directly relevant to understanding how the various scans performed over the entire patient cohort and the supplementary materials giving readers a more robust insight into the actual image quality than any particular metric likely ever could. Revised figures below:

Figure 6: Image quality metrics. Note that “better” performance by each metric is higher CNR, lower TIW.

Scan Characteristics

Scan Name	Reconstruction Method	Number of Projections	Mean Scan Time (s)	Data Structure Accuracy (°)
Conventional	Conventional	1,320	240	N/A
STO600	Conventional	600	242	0.71
STO200	Motion Compensated	200	91	1.75

Image Quality Quantification

Scan Name	CNR	TIW-T (mm)	TIW-D (mm)
Conventional	7.5	7.8	7.7
STO600	5.9	10.2	9.4
STO200	12.4	5	3.5

Table 1: Summary of the acquired scans and reconstructed image quality.

Co-reviewed by
Jamie McClelland and Yuliang Huang

Thank you both for the thorough review. I appreciate the concerns raised and believe that with the substantial supplementary materials the readers will be able to make a more thorough evaluation of the key aspects of the work, while the quantitative metrics allow them to extrapolate how those qualitative evaluation might extend over the entire cohort. I hope you agree that the manuscript being submitted is a thorough, extensive and robust representation of the work and will be of interest to the journal readership.